# A modified BCG with depletion of enzymes associated with peptidoglycan amidation induces enhanced protection against tuberculosis in mice

Moagi Tube Shaku[1,2], Peter K Um[2], Karl L Ocius[3], Alexis J Apostolos[3], Marcos M Pires[3], William R Bishai[2], Bavesh D Kana[1]*

[1]DST/NRF Centre of Excellence for Biomedical TB Research, Faculty of Health Sciences, University of the Witwatersrand, National Health Laboratory Service, Johannesburg, South Africa; [2]Center for Tuberculosis Research, Department of Medicine, Johns Hopkins School of Medicine, Baltimore, United States; [3]Department of Chemistry, University of Virginia, Charlottesville, United States

**Abstract** Mechanisms by which *Mycobacterium tuberculosis* (Mtb) evades pathogen recognition receptor activation during infection may offer insights for the development of improved tuberculosis (TB) vaccines. Whilst Mtb elicits NOD-2 activation through host recognition of its peptidoglycan-derived muramyl dipeptide (MDP), it masks the endogenous NOD-1 ligand through amidation of glutamate at the second position in peptidoglycan side-chains. As the current BCG vaccine is derived from pathogenic mycobacteria, a similar situation prevails. To alleviate this masking ability and to potentially improve efficacy of the BCG vaccine, we used CRISPRi to inhibit expression of the essential enzyme pair, MurT-GatD, implicated in amidation of peptidoglycan side-chains. We demonstrate that depletion of these enzymes results in reduced growth, cell wall defects, increased susceptibility to antibiotics, altered spatial localization of new peptidoglycan and increased NOD-1 expression in macrophages. In cell culture experiments, training of a human monocyte cell line with this recombinant BCG yielded improved control of Mtb growth. In the murine model of TB infection, we demonstrate that depletion of MurT-GatD in BCG, which is expected to unmask the D-glutamate diaminopimelate (iE-DAP) NOD-1 ligand, yields superior prevention of TB disease compared to the standard BCG vaccine. *In vitro* and *in vivo* experiments in this study demonstrate the feasibility of gene regulation platforms such as CRISPRi to alter antigen presentation in BCG in a bespoke manner that tunes immunity towards more effective protection against TB disease.

## Editor's evaluation

This important study provides evidence for a new target to improve vaccination against tuberculosis. The authors provide compelling evidence that inactivation of an essential enzyme pair in Mycobacterium bovis BCG, the only licensed vaccine against tuberculosis, enhances protection in a mouse model of the disease. The work will be of interest to researchers working on tuberculosis vaccine development.

## Introduction

Tuberculosis (TB) caused by *Mycobacterium tuberculosis* (Mtb) remains a leading cause of death from an infectious disease worldwide (***WHO, 2022***). Despite the availability of the Bacille Calmette Guerin

*For correspondence:
bavesh.kana@nhls.ac.za

**eLife digest** Tuberculosis is the leading cause of death from an infectious disease worldwide, partially due to a lack of access to drug treatments in certain countries where the disease is common. The only available tuberculosis vaccine – known as the BCG vaccine – is useful for preventing cases in young children, but is ineffective in teenagers and adults. So, there is a need to develop new vaccines that offer better, and longer lasting, durable protection in people of all ages.

During an infection, our immune system recognizes markers known as PAMPs on the surface of bacteria, viruses or other disease-causing pathogens. The recognition of PAMPs by the immune system enables the body to distinguish foreign invading organisms from its own cells and tissues, thus triggering a response that fights the infection. If the body encounters the infectious agent again in the future, the immune system is able to quickly recognize and eliminate it before it can cause disease. Vaccines protect us by mimicking the appearance of the pathogen to trigger the first immune response without causing the illness.

The BCG vaccine contains live bacteria that are closely related to the bacterium responsible for tuberculosis called *Mycobacterium tuberculosis*. Both *M. tuberculosis* and the live bacteria used in the BCG vaccine are able to hide an important PAMP, known as the NOD-1 ligand, from the immune system, making it harder for the body to detect them. The NOD-1 ligand forms part of the bacterial cell wall and modifying the BCG bacterium so it cannot disguise this PAMP may lead to a new, more effective vaccine.

To investigate this possibility, Shaku et al. used a gene editing approach to develop a modified version of the BCG bacterium which is unable to hide its NOD-1 ligand when treated with a specific drug. Immune cells trained with the modified BCG vaccine were more effective at controlling the growth of *M. tuberculosis* than macrophages trained using the original vaccine. Furthermore, mice vaccinated with the modified BCG vaccine were better able to limit *M. tuberculosis* growth in their lungs than mice that had received the original vaccine.

These findings offer a new candidate vaccine in the fight against tuberculosis. Further studies will be needed to modify the vaccine for use in humans. More broadly, this work demonstrates that gene editing can be used to expose a specific PAMP present in a live vaccine. This may help develop more effective vaccines for other diseases in the future.

(BCG) TB vaccine, approximately 2 billion people worldwide are latently infected with Mtb and represent a reservoir of future active disease (*Trunz et al., 2006*). BCG is the only licensed TB vaccine and has been in use since the 1920 s, with close to 100 million infants vaccinated annually worldwide (*Trunz et al., 2006*). BCG protects against TB meningitis and miliary TB in children, but lacks efficacy against pulmonary TB in adults (*Martinez et al., 2022*); hence, improved TB vaccines remain an urgent public health priority.

Innate immune pattern recognition receptors (PRRs) have evolved to sense unique pathogen-associated molecular patterns (PAMPs) that are often essential components of infecting organisms (*Li and Wu, 2021*). Bacterial peptidoglycan (PG) is one such PAMP, and it is detected by the PRRs NOD-1, which recognizes the D-isoglutamate diaminopimelate (iE-DAP) segment of PG, and NOD-2 which detects the related muramyl dipeptide (MDP) portion of PG (*Li and Wu, 2021*). Activation of NOD-1 triggers the production of pro-inflammatory cytokines through nuclear factor κB (NF-κB) and mitogen-activated protein kinase (MAPK) pathways and similarly, NOD-2 activation leads to upregulation of NF-κB activity (*Caruso et al., 2014*).

While many gram-negative pathogens express abundant levels of the NOD-1 ligand iE-DAP (*Caruso et al., 2014*), pathogenic mycobacteria including Mtb, *M. bovis*, and the *M. bovis*-derived BCG strains possess an immune subversion system which enzymatically masks NOD-1 antigenic structure through amidation, thereby enabling escape from NOD-1 mediated immune containment (*Maitra et al., 2019*). The enzyme pair encoded by the *murT* (Mb3739, in *M. bovis*)-*gatD* (Mb3740) operon forms a glutaminase and an amidotransferase complex, which amidates iE-DAP to form iQ-DAP, thus avoiding NOD-1 detection (*Figure 1—figure supplement 1*). As amidation of D-isoglutamate to D-isoglutamine during PG maturation in mycobacteria is required for subsequent PG cross-linking, MurT and GatD are important for mycobacterial cell wall integrity and genetic

screens have confirmed their essentiality for *in vitro* survival (*de Wet et al., 2020*; *Maitra et al., 2021*; *Shaku et al., 2023*).

We hypothesized that depletion of MurT-GatD in BCG would result in increased abundance of the NOD-1 ligand (iE-DAP), thus enabling enhanced immunogenicity of the recombinant vaccine strain. We used a CRISPRi platform for targeted inhibition of transcription of the amidotransferase complex - MurT-GatD essential for PG amidation in mycobacteria (i.e. modification of iE-DAP to iQ-DAP) to develop a recombinant BCG vaccine (rBCG::iE-DAP) engineered to activate NOD-1 during vaccination. CRISPRi mediated genetic manipulation showed that MurT-GatD levels can be conditionally depleted in BCG without complete loss of viability. Compared to the wildtype (WT) BCG, vaccination of mice with the MurT-GatD-depleted rBCG gives superior containment of Mtb proliferation in lungs.

## Results
### Construction of rBCG::iE-DAP
Using the CRISPRi gene expression knockdown system, we generated a derivative of plasmid pLRJ965 (*Rock et al., 2017*) that conditionally expresses dCas9 from *Streptococcus thermophiles* and a 17 base short guide RNA (sgRNA) sequence that targets the *murT-gatD* operon upon exposure to anhydrotetracycline (ATc) or doxycycline (Dox) to create plasmid PLRJ965 +*murT*sgRNA (*Figure 1—figure supplement 2a and b*). This plasmid was introduced into BCG-Pasteur to generate a recombinant BCG strain called rBCG::iE-DAP. We showed that following ATc induction, the relative mRNA levels of the full length *murT* transcript were 1000-fold lower in rBCG::iE-DAP when compared with the uninduced rBCG strain (*Figure 1b*).

Next, we evaluated the impact of MurT-GatD depletion on BCG viability. CRISPRi mediated inhibition of *murT-gatD* transcription in rBCG::iE-DAP by supplementation of growth media with an increasing concentration of ATc [0–500 ng/ml] resulted in growth inhibition of the recombinant strain (*Figure 1—figure supplement 2c*). This is consistent with earlier knockdown of a MurT homologue in *Mycobacterium smegmatis* (MSMEG_6276) which revealed a growth defect upon CRISPRi mediated MSMEG_6276 depletion (*Shaku et al., 2023*).

### MurT-GatD depletion in BCG causes expression of the NOD-1 ligand (iE-DAP) and increased NOD-1 signaling
To determine the effects of MurT-GatD depletion in BCG, we performed scanning and transmission electron microscopy (SEM, TEM). As shown in *Figure 1c*, SEM revealed a well-formed typical mycobacterial outer cell wall structure in WT BCG, whereas rBCG::iE-DAP cells displayed a wrinkled outer-cell wall structure, sometimes with indentations. Quantification of SEM fields revealed a 70% increase in the frequency of bacilli with these defects in rBCG::iE-DAP relative to the WT BCG (*Figure 1d*). Consistent with this, TEM revealed a typical multi-layered mycobacterial cell wall outline (*Mahapatra et al., 2008*), with visible layers in WT BCG in comparison to the defective cell wall structure in rBCG::iE-DAP, without a clear cell wall outline as shown in *Figure 1—figure supplement 3*. Upon counting individual cells in TEM fields, we observed a 65% increase in the frequency of wall defects in rBCG::iE-DAP compared with the WT BCG strain (*Figure 1f*).

We hypothesized that reduced PG cross-linking due to MurT-GatD depletion, and the concomitant cell wall defects, might potentiate cell wall targeting antibiotics in rBCG::iE-DAP compared to WT BCG. Indeed, as shown in *Table 1*, MurT-GatD knockdown was associated with a 2- to 16-fold decrease in the minimal inhibitory concentrations of the recombinant strain for amoxicillin-clavulanate, meropenem, vancomycin, and ethionamide, each of which targets either PG biosynthesis or PG-dependent accessory glycolipids. To further confirm the reduced levels of PG cross-linking, we stained MurT-GatD depleted rBCG::iE-DAP cells with BODIPY-FL vancomycin—a fluorescent probe which specifically labels uncrosslinked PG. As shown in the confocal fluorescence micrographs in *Figure 1g*, BODIPY-FL vancomycin displayed complete cell wall labeling of the rBCG::iE-DAP cells, in contrast, only the poles of WT BCG cells were labeled. This corresponds to the known polar elongation of BCG cells and the relative abundance of new, uncross-linked PG at the cell poles (*Aldridge et al., 2012*; *Joyce et al., 2012*).

To specifically demonstrate that MurT-GatD depletion resulted in reduced amidation of iE-DAP, we used a fluorogenic amidated, synthetic tetrapeptide, TetraFl (TAMRA fluorophore-L-Ala-D-Gln-L-Lys-D-Ala).

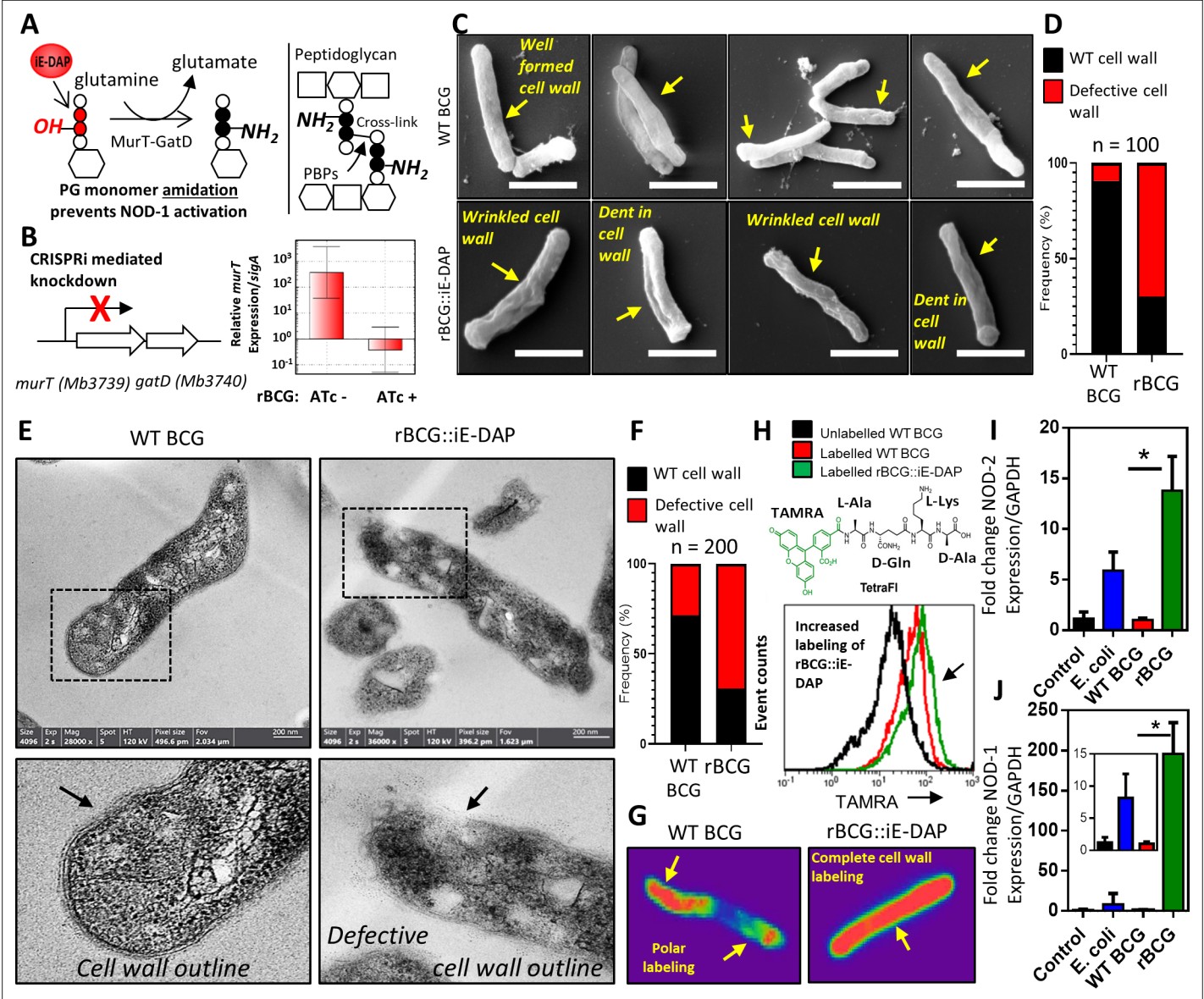

**Figure 1.** Phenotypic characterization of rBCG::iE-DAP and NOD-1 activation. (**A**) Schematic representation of MurT-GatD mediated PG precursor amidation. (**B**) *murT* gene expression measured by quantitative PCR in rBCG::iE-DAP. (**C**) Scanning electron micrographs of WT BCG (n=45 micrographs, 100 cells counted) and rBCG::iE-DAP (n=48 micrographs, 100 cells counted) grown in media supplemented with 200 ng/ml ATc. Scale bar = 1 µm. (**D**) Frequency of cells with cell wall defects as seen by SEM. (**E**) Transmission electron micrographs of WT BCG (n=45 micrographs, 200 cells counted) and rBCG::iE-DAP (n=45 micrographs, 200 cells counted) grown in media supplemented with 200 ng/ml ATc. Scale bar = 200 nm. (**F**) Frequency of cells with cell wall defects as seen by TEM. (**G**) MurT-GatD depleted cells labeled with fluorescent BODIPY-FL vancomycin. (**H**) Flow cytometry analysis of WT BCG and rBCG::iE-DAP cells labelled with a PG amidation reporter probe TAMRA-L-Ala-D-glutamine-L-Lys-D-Ala (TetraFl). (**I**) *nod-2* gene expression measured by quantitative PCR in INFγ activated THP-1 macrophages stimulated with *E. coli*, WT BCG and rBCG::iE-DAP. (**J**) *nod-1* gene expression measured by quantitative PCR in INFγ activated THP-1 macrophages infected with *E. coli*, WT BCG and rBCG::iE-DAP. Three independent biological repeats (n=3) were assessed. Student *t*-test was used for statistical analysis. The error bars represent the standard deviation relative to the mean. *: p-value <0.01.

The online version of this article includes the following source data and figure supplement(s) for figure 1:

**Source data 1.** Phenotypic characterization of rBCG::iE-DAP and NOD-1 activation.

**Figure supplement 1.** Molecular structures of iE-DAP and iQ-DAP.

**Figure supplement 2.** CRISPRi depletion of MurT-GatD in rBCG::iE-DAP.

**Figure supplement 2—source data 1.** Growth kinetics of rBCG:: iE-DAP.

**Figure supplement 3.** TEM reveals defective cell wall of rBCG::iE-DAP.

*Figure 1 continued on next page*

*Figure 1 continued*

**Figure supplement 4.** Depletion of MurT and GatD causes reduced PG amidation.

**Figure supplement 4—source data 1.** Depletion of MurT and GatD causes reduced PG amidation.

**Figure supplement 5.** qPCR of *nod-1* and *nod-2* expression in non-activated THP-1 macrophages.

**Figure supplement 5—source data 1.** qPCR of *nod-1* and *nod-2* expression in THP-1 macrophages.

This amidated, D-Gln-containing probe is incorporated into mycobacterial PG by the activity of PG cross-linking L,D-transpeptidases which require the amidation modification on one of the PG stem peptides to form the cross-link (*Pidgeon et al., 2019*). The deficient cross-linking in the cell wall due to MurT-GatD depletion led us to speculate that more of the amidated probe will be incorporated into existing PG. As seen in *Figure 1h*, labeling of the MurT-GatD depleted cells showed a greater incorporation of the tetrapeptide fluorophore than in WT BCG. We further assessed this reduced amidation by labeling of PG extracted from WT BCG and rBCG::iE-DAP with an amine reactive fluorescent dye, which binds amine ($NH_2$) groups. This revealed decreased amidation of PG upon MurT-GatD depletion (*Figure 1—figure supplement 4*), thus confirming that rBCG::iE-DAP displayed greater exposure of the iE-DAP, NOD-1 antigenic structure. To investigate the involvement of NOD-1 in macrophages, we next assessed the ability of rBCG::iE-DAP to induce increased expression of the NOD PRRs by infecting interferon-gamma (IFNγ) activated THP-1 macrophages at an MOI of 1. This was followed by quantitative PCR (qPCR) to measure expression of both *nod-1* and *nod-2* in comparison to the wildtype parental strain (WT BCG) 12 hr post infection. We also infected the macrophages with *E. coli*, which naturally expresses the NOD-1 ligand iE-DAP in its PG (*Girardin et al., 2003*). This experiment was also performed in non-activated THP-1 macrophages. As shown in *Figure 1—figure supplement 5*, *nod-1* and *nod-2* expression was present at basal levels in non-activated and uninfected cells as previously shown (*Juárez et al., 2012*; *Rommereim et al., 2020*); WT BCG infection did not induce significant changes in *nod-1/2* expression and *E. coli* infection led to increased expression of both NOD receptors in both IFNγ activated and non-activated macrophages, as expected. Infection with rBCG::iE-DAP led to an ~15-fold increase in NOD-2 expression in IFNγ-activated macrophages, an ~55-fold and ~200-fold increase in *nod-1* expression in both non-activated and IFNγ-activated THP-1 macrophages, respectively (*Figure 1i, j*). These differences in gene expression were significantly higher than those noted for the parental BCG strain.

## rBCG::iE-DAP is responsive to anhydrotetracycline activation *in vivo* and causes increased TNFα expression in bone marrow derived macrophages (BMDMs)

To test the hypothesis that inhibition of MurT-GatD expression in rBCG::iE-DAP enhances the immunogenicity of the recombinant strain, we first infected IFNγ-activated bone marrow derived macrophages (BMDMs) with rBCG::iE-DAP and supplemented the growth media with increasing concentrations of ATc. This was done to assess activation of the CRISPRi system *ex vivo* and also to compare growth to WT BCG infected cells. The growth of the strains was recorded by plating for colony forming unit (CFU) counts at day 3 and day 5 post-infection. At day 3, bacterial containment was observed for all strains but was most prominent for rBCG::iE-DAP strains treated with ATc. Dose-dependent inhibition

**Table 1.** Minimum inhibitory concentrations of cell wall targeting antibiotics on rBCG::iE-DAP.

| Drug (μg/ml) | WT BCG (MIC) | rBCG (MIC) |
| --- | --- | --- |
| Amoxicillin | >64 | >64 |
| Amoxicillin-Clavulanate | >64 | 8 |
| Meropenem | 32 | 2 |
| Vancomycin | 8 | 4 |
| Ethionamide | >64 | 32 |

The online version of this article includes the following source data for table 1:

**Source data 1.** Minimum inhibitory concentrations of cell wall targeting antibiotics.

of growth of rBCG::iE-DAP was observed at day 5, with 500 ng/ml ATc (the maximum concentration used) resulting in an ~three fold difference in growth inhibition of rBCG::iE-DAP in comparison to WT BCG and rBCG::iE-DAP without ATc supplementation (*Figure 2a*). Secondly, we performed ELISA experiments to assess the expression of the pro-inflammatory cytokine TNFα as rBCG::iE-DAP is designed to express the NOD-1 ligand iE-DAP, potentially increasing the pro-inflammatory response. Activation of rBCG::iE-DAP by supplementation of growth media with ATc resulted in a dose-dependent increase in TNFα expression in comparison to WT BCG in IFNγ-activated BMDMs. However, this was statistically insignificant between strains, while TNFα expression remained low for both WT BCG and rBCG::iE-DAP strains when used for infection of unactivated BMDMs (*Figure 2— figure supplement 1*). These results demonstrate that rBCG::iE-DAP is responsive to activation *ex vivo* and can be tested *in vivo*.

## rBCG::iE-DAP *in vitro* trained macrophages control Mtb H37Rv growth

WT BCG trains macrophages in a NOD-2 dependent manner and as a result, killing of Mtb is enhanced if the trained cells are exposed to Mtb at a later stage (*Kleinnijenhuis et al., 2012*; *Kaufmann et al., 2018*). We hypothesized that rBCG::iE-DAP engineered to express the NOD-1 ligand upon activation with ATc will lead to enhanced macrophage training activity, resulting in better control of Mtb growth compared to WT BCG trained macrophages. To test this, we used an *in vitro* macrophage training assay to assess the Mtb killing ability of rBCG::iE-DAP trained macrophages. Lipopolysaccharide (LPS) and murein dipeptide (MDP) were used as controls and a cells-only (RPMI) control was also included. LPS activates toll like receptor (TLR)–4 leading to monocyte activation (*Fujihara et al., 2003*) and MDP activates NOD-2 leading to macrophage training (*van der Heijden et al., 2018*). As shown in *Figure 2b*, heat-killed rBCG::iE-DAP trained macrophages displayed increased control of Mtb H37Rv compared to heat-killed WT BCG trained macrophages and MDP trained macrophage. Macrophages derived from LPS stimulated monocytes did not control Mtb growth. Based on these promising findings, we proceeded to test rBCG::iE-DAP in the murine model of TB infection.

## rBCG::iE-DAP activation *in vitro* and in mice-aerosol infections with doxycycline

Doxycycline (Dox), a tetracycline analog, is used in *in vivo* TB models for temporal regulation of mycobacterial gene expression (*Miow et al., 2021*). The CRISPRi platform used for generation of rBCG::iE-DAP is also based on a Dox-responsive TetR-*tetO* unit which in the presence of doxycycline leads to expression of the CRISPRi system and subsequent transcriptional inhibition of *murT-gatD* (*Rock et al., 2017*). To assess the activation of rBCG::iE-DAP with Dox, the strain was grown in an increasing range of Dox concentrations to assess the activation of CRISPRi *in vitro*, a WT BCG+Dox control experiment was also included. Activation of CRISPRi in rBCG::iE-DAP with Dox resulted in a dose-dependent reduction of rBCG::iE-DAP growth (*Figure 2c, d*), which was corroborated when growth was assessed by CFU counts while WT BCG was not affected by Dox supplementation (*Figure 2c, d*).

To test the activation of rBCG::iE-DAP *in vivo* and to determine the minimum effective dose of Dox, we aerosol infected BALB/c mice with ~100 CFU of rBCG::iE-DAP and administered Dox for 10 days at doses ranging from 0.125 to 1 mg/kg/day by oral gavage (*Figure 2e*). Administration of 1 mg/kg/day resulted in a significant reduction in growth of rBCG::iE-DAP in the lungs of the mice (*Figure 2f*). We further assessed long-term retention of activation of rBCG::iE-DAP by performing the activation experiment for 4 weeks and this revealed long-term activation of rBCG::iE-DAP and retention of the CRISPRi plasmid (PLRJ965 +*murT*sgRNA, which has a kanamycin [Kan] resistance cassette) by rBCG::iE-DAP *in vivo* (*Figure 2—figure supplement 2*). We also plated lung homogenates on media containing Kan and found that recovered rBCG::iE-DAP bacilli formed similar CFU counts on media with or without Kan and PCR amplification of the dCas9 allele from recovered rBCG::iE-DAP bacilli revealed retention of the CRISPRi plasmid and the bacilli were responsive to ATc treatment (*Figure 2f, g*). Similarly, as shown in *Figure 2h, i*, at 8 weeks post infection, rBCG::iE-DAP bacilli recovered from the lungs of infected mice formed small colonies on solid agar in comparison to recovered WT BCG bacilli, indicative of the long-term efficacy of 1 mg/kg/day Dox *in vivo* for CRISPRi activation. These results demonstrate retention of the CRISPRi plasmid by rBCG::iE-DAP *in vivo*.

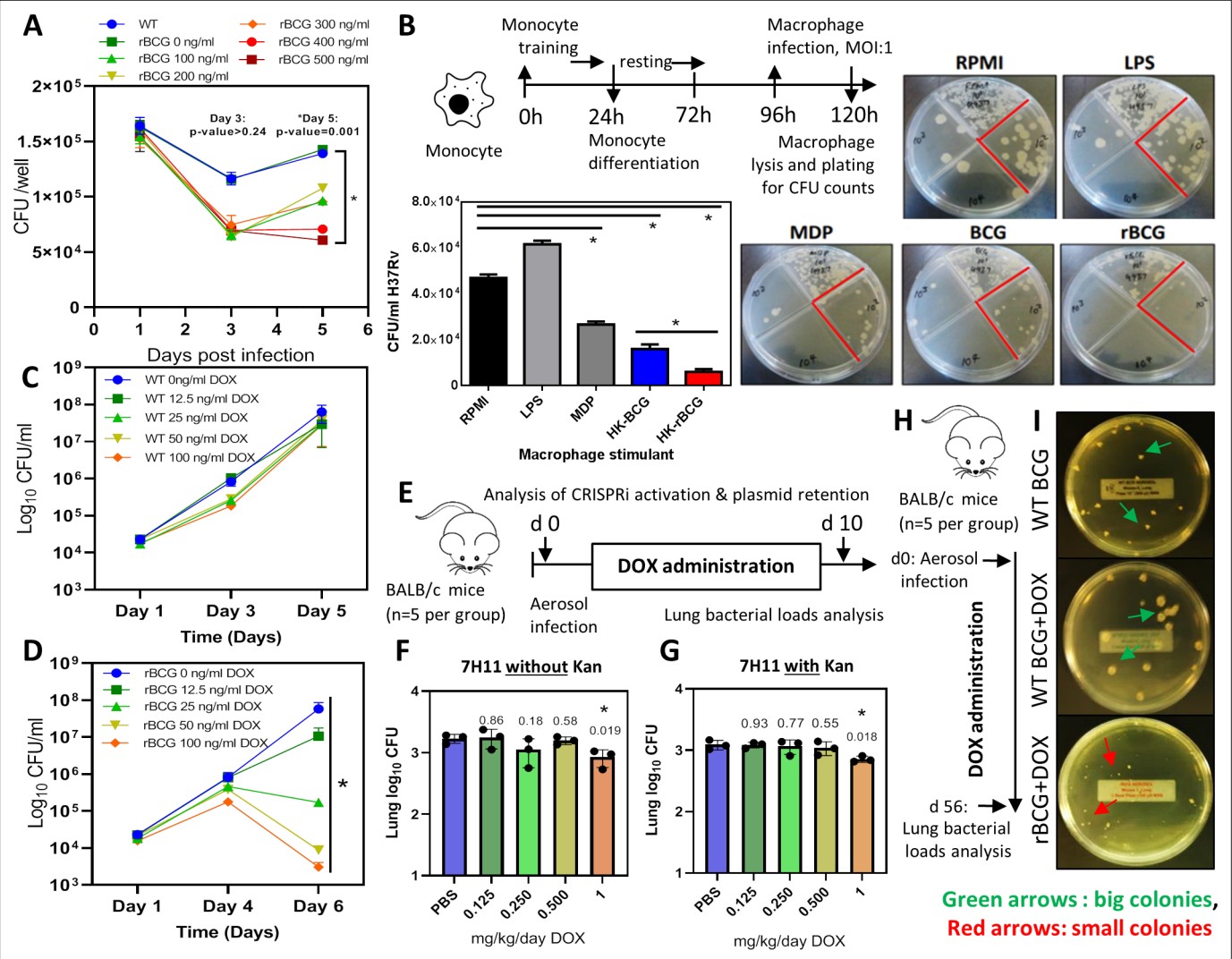

**Figure 2.** Survival of rBCG::iE-DAP in IFNγ activated bone marrow derived macrophages (BMDMs), training of monocytes and activation with doxycycline. (**A**) IFNγ-activated BMDMs (1x10⁶ cells) were infected at MOI: 1 with WT BCG and rBCG::iE-DAP. ATc was added to culture media for induction of the CRISPRi system in rBCG::iE-DAP at concentrations ranging from 100 ng/ml – 500 ng/ml and growth of the strains was assessed after 3 and 5 days. (**B**) Training of U937 monocytes with heat-killed (HK)-rBCG::iE-DAP compared to HK-WT BCG. Shown is also the representative plates for the experiment. (**C, D**) CFU counts of *in vitro* grown WT BCG and of rBCG::iE-DAP grown in complete 7H9 medium at varying concentrations of Dox. (**E**) Determination of the Dox concentration for activation of rBCG::iE-DAP *in vivo*. Mice were aerosol infected with ~2.5 log10 CFU of rBCG and Dox (0.125–1 mg/kg/day) - was administered by oral gavage for 10 days. (**F, G**) CFU counts from the experiment shown in panel E. Lung homogenates were plated on both 7H11 with (**G**) and without (**F**) kanamycin (25 µg/ml) to assess the loss of the CRISPRi plasmid during *in vivo* growth. p-values are given above the graphs. (**H**) Aerosol infection of mice with ~2.5 log10 CFU of WT BCG, rBCG::iE-DAP and administration of Dox (1 mg/kg/day) for 8 weeks. (**I**) Plates showing the colony size of rBCG::iE-DAP+Dox compared to WT BCG or WT BCG+Dox, recovered from the lungs of aerosol infected mice from the experiment shown in panel H. Three independent biological repeats (n=3) were assessed for the *in vitro* experiments, the error bars represent the standard deviation relative to the mean. Five mice per group (n=5) were used for the *in vivo* experiments. Student *t*-test was used for statistical analysis. The error bars represent the standard deviation relative to the mean. *: p-value <0.05.

The online version of this article includes the following source data and figure supplement(s) for figure 2:

**Source data 1.** Activation of rBCG::iE-DAP in BMDMs, training of monocytes and activation with doxycycline.

**Figure supplement 1.** Analysis of secreted TNFα levels from non-activated and IFN-activated BMDMs infected with WT BCG and rBCG::iE-DAP at MOI 1:20.

**Figure supplement 1—source data 1.** TNFα ELISA of non-activated and IFNγ-activated BMDMs infected with WT BCG and rBCG:: iE-DAP.

**Figure supplement 2.** Efficacy of 1 mg/kg/day dose of doxycycline for CRISPRi activation.

**Figure supplement 2—source data 1.** Efficacy of 1 mg/kg/day dose of doxycycline for CRISPRi *Figure 2—figure supplement 2d* activation.

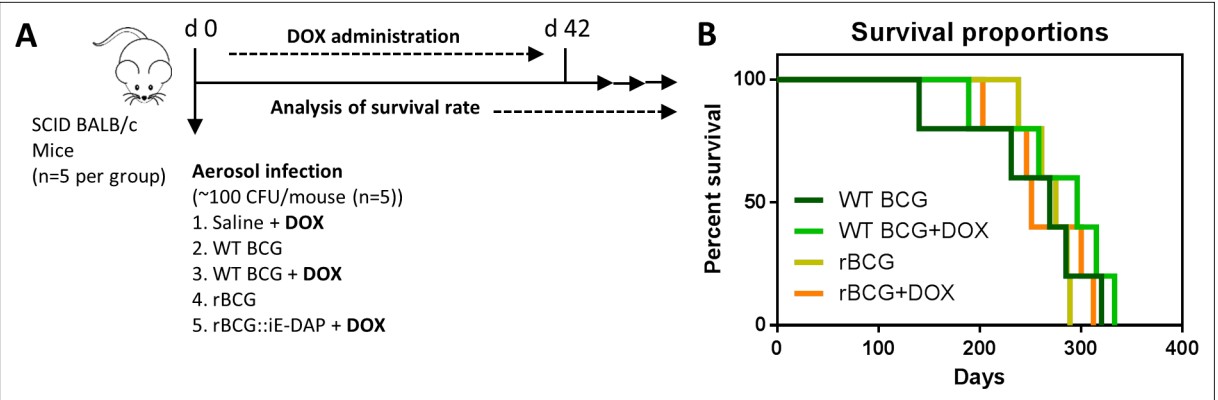

**Figure 3.** Analysis of rBCG::iE-DAP strain attenuation. (**A**) Schematic representation of SCID mice aerosol infection with WT BCG and rBCG::iE-DAP for analysis of strain attenuation. rBCG::iE-DAP activation *in vivo* was performed by administration of Dox at 1 mg/kg/day. SCID mice (n=5 per group) were aerosol infected with ~2.5 log10 CFU of WT BCG or rBCG::iE-DAP, a WT BCG+Dox group was included as a control. (**B**) Percent survival of SCID mice following low-dose challenge with WT BCG and rBCG compared to WT BCG+Dox or rBCG+Dox groups. Five mice per group (n=5) were used for the *in vivo* experiments. student *t*-test was used for statistical analysis.

The online version of this article includes the following source data for figure 3:

**Source data 1.** Analysis of rBCG::iE-DAP strain attenuation.

## Analysis of rBCG::iE-DAP attenuation in SCID mice

To further explore the phenotype and attenuation of rBCG::iE-DAP *in vivo*, we aerosol infected female SCID (severe combined immunodeficiency) mice with a low dose (~100 CFU) of WT BCG and rBCG::iE-DAP, and included Dox receiving groups (i.e.WT BCG+Dox and rBCG::iE-DAP+Dox) (*Figure 3a*). WT BCG infected mice displayed early decreased survival as expected, followed by the WT BCG+Dox group of mice (*Figure 3b*). Although rBCG::iE-DAP infected mice, either receiving Dox or not, displayed slight increased survival, this data was not significantly different from WT BCG-infected mice. This suggested that rBCG::iE-DAP upon CRISPRi activation is not more attenuated than WT BCG and does not cause more disease in SCID mice compared to WT BCG.

## rBCG::iE-DAP induces enhanced protection against *Mycobacterium tuberculosis* infection in mice compared to WT BCG

To assess the protective efficacy of rBCG::iE-DAP against TB infection relative to the standard BCG vaccine, we immunized groups of BALB/c mice (n=5 per group) intradermally with WT BCG or rBCG::iE-DAP (*Figure 4a*). rBCG::iE-DAP immunized mice received a Dox dose by oral gavage at 1 mg/kg/day for activation of CRISPRi *in vivo* and we also included a Saline+Dox group, a WT-BCG+Dox group and a rBCG::iE-DAP without Dox group as controls for the vaccination experiment. The immunized mice receiving Dox were weighed weekly for 6 weeks prior to Mtb challenge to assess the effect of daily Dox administration on the health of the mice (*Figure 4a*). We assessed the percentage weight change of mice receiving Dox relative to the no-Dox groups and found that the weights of the different groups remained within 80–100% of baseline with few non-significant differences at week 6 (*Figure 4b*).

After 6 weeks, the immunized mice were challenged with ~100 CFU of Mtb H37Rv via the aerosol route and mycobacterial loads were determined in lungs and spleens at 4 and 8 weeks post challenge (*Figure 4a*). At 4 weeks post Mtb challenge, mice were sacrificed to assess lung pathology and bacterial burden in the lungs and spleens. As seen in *Figure 4—figure supplement 1a*, at week 4 post-infection, the WT BCG+Dox group displayed significantly lower lung weights compared to the Saline+Dox group, while the WT BCG without Dox-treatment and the rBCG::iE-DAP with or without Dox-treatment groups displayed similar lung weights compared to the Saline+Dox group. Similarly as shown in *Figure 4—figure supplement 1b*, the WT BCG+Dox group displayed significantly lower spleen weights compared to the Saline+Dox group while the WT BCG without Dox-treatment and the rBCG::iE-DAP with or without Dox-treatment groups displayed similar spleen weights compared to the Saline+Dox group. Analysis of lung and spleen bacterial burdens at 4 weeks post infection

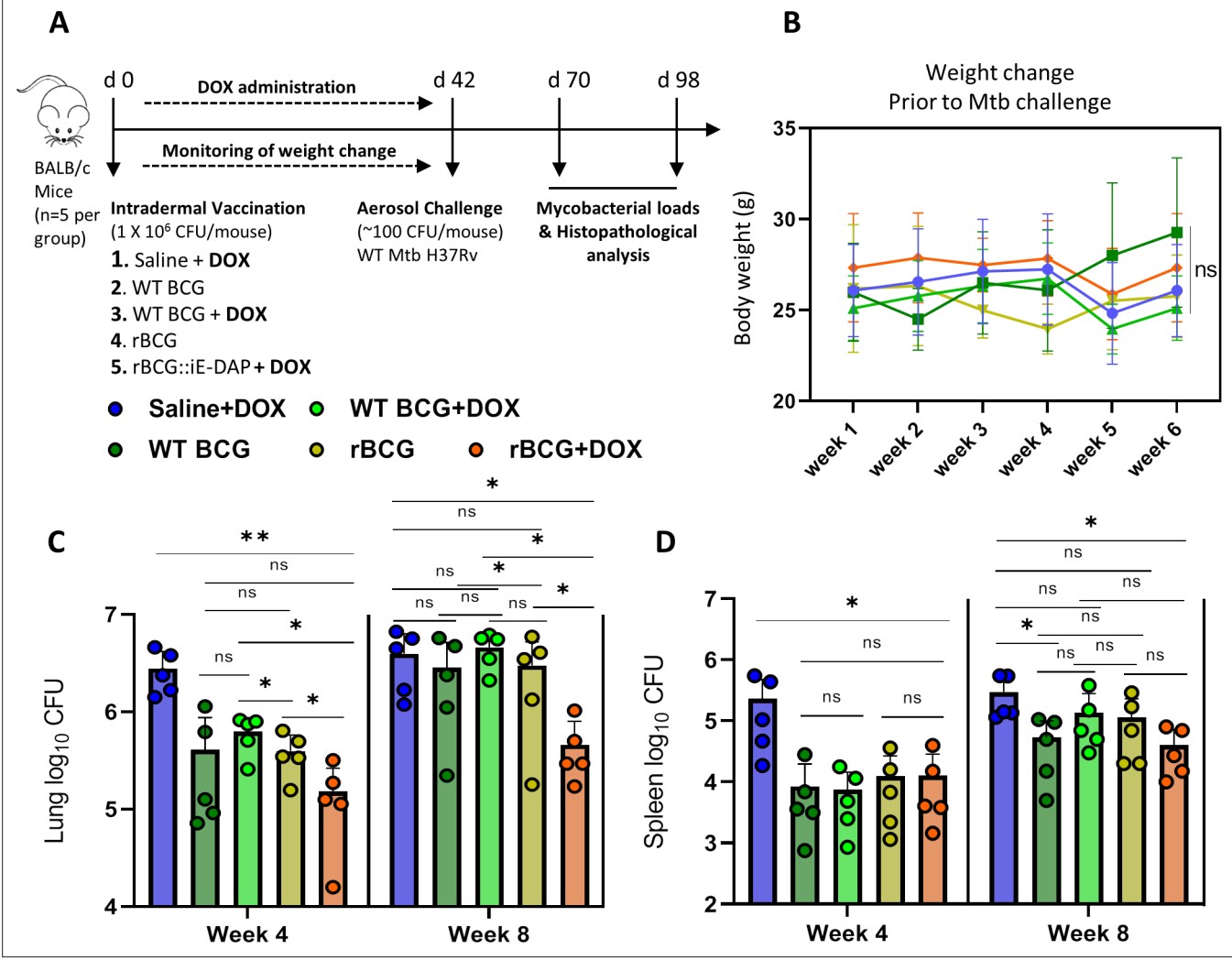

**Figure 4.** Efficacy of rBCG::iE-DAP in comparison to standard WT BCG for protection against Mtb H37Rv infection in mice. (**A**) Schematic representation of the mouse immunization and Mtb H37Rv challenge protocol. (**B**) Percentage weight change at week 6 (day 42) immediately prior to Mtb challenge. (**C, D**) Lung and Spleen bacterial burdens at week 4 and week 8 post-challenge with Mtb. Five mice per group (n=5) were used for the *in vivo* experiments. Student *t*-test was used for statistical analysis. The error bars represent the standard deviation relative to the mean. *: p-value <0.05, **: p-value <0.01.

The online version of this article includes the following source data and figure supplement(s) for figure 4:

**Source data 1.** Efficacy of rBCG::iE-DAP in comparison to standard WT BCG for protection against Mtb H37Rv infection in mice.

**Figure supplement 1.** Lumg and spleen weights post-challenge with Mtb.

**Figure supplement 1—source data 1.** Lung and spleen weights post-challenge with Mtb.

revealed that rBCG::iE-DAP+Dox was superior to WT BCG and WT BCG+Dox in protecting against Mtb challenge in the lungs and reduced bacterial burden in the spleen similar to WT BCG or WT BCG+Dox. Also as seen in *Figure 4—figure supplement 1a* at 8 weeks post Mtb challenge, the rBCG::iE-DAP+Dox group displayed reduced lung weights compared to the Saline+Dox group and the WT BCG vaccinated group indicative of control of bacterial burden and indeed, analysis of lung bacterial burden corroborated findings at 4 weeks that rBCG::iE-DAP+Dox was superior to WT BCG or WT BCG+Dox in controlling Mtb growth in the lung (*Figure 4c*). At week 8, WT BCG or WT BCG+Dox vaccination both displayed waning efficacy in this model, as previously shown (*Henao-Tamayo et al., 2015*; *Dwivedi et al., 2022*). In the spleen, rBCG::iE-DAP+Dox displayed similar efficacy to WT BCG or WT BCG+Dox for control of infection compared to the Saline+Dox group (*Figure 4d*).

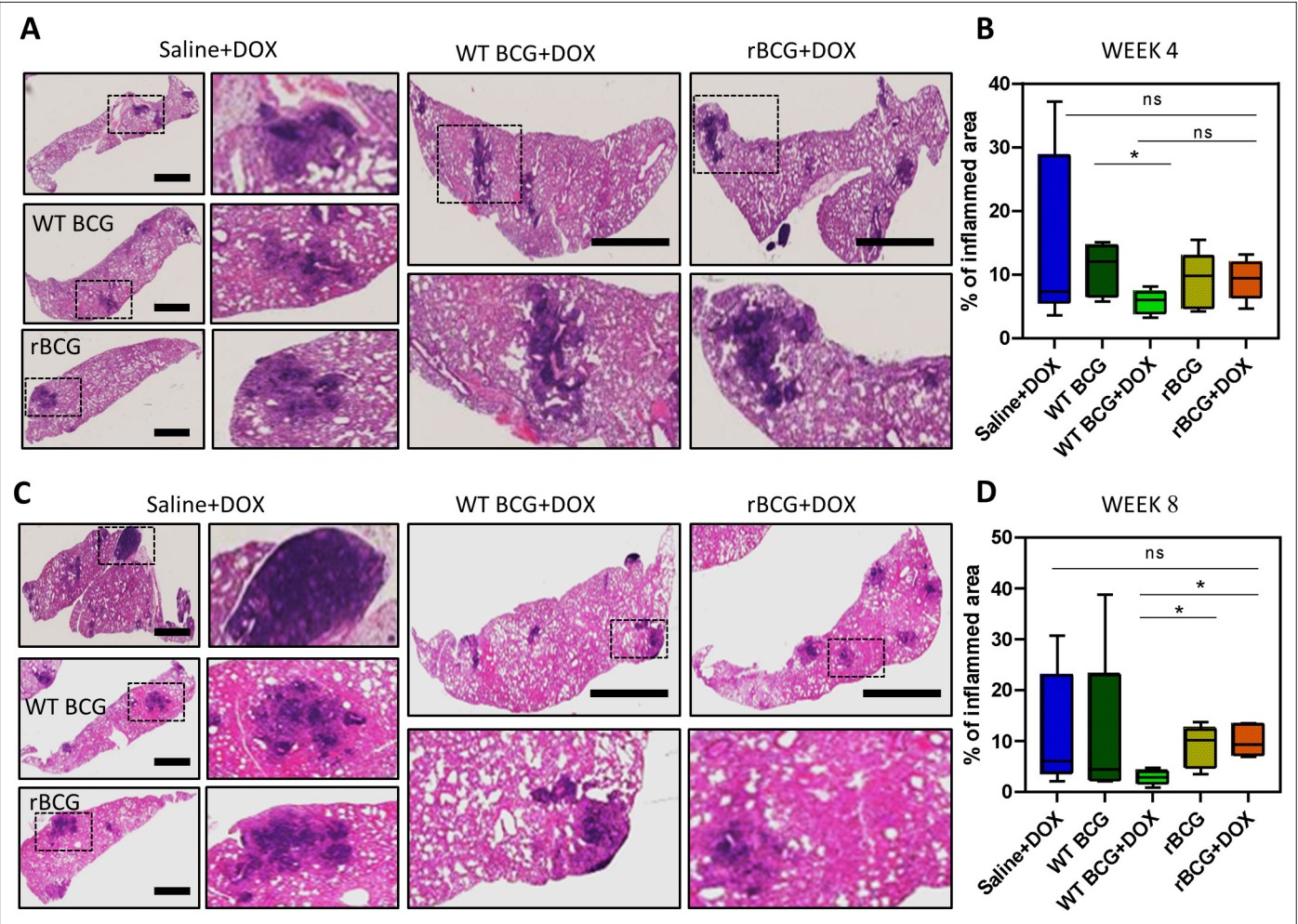

**Figure 5.** Histopathological analysis of lung samples. (**A**) Histological haematoxylin and eosin (H&E) staining of lung samples at week 4 post Mtb challenge. Scale bar = 2.5 mm. (**B**) Analysis of percentage of inflamed area (indicated with black boxes) from each mouse lung per immunized group (n=5 per group), shows that rBCG::iE-DAP+Dox immunized mice present with early lung inflammation compared to WT BCG+Dox. (**C**) H&E staining of lung samples at week 8 post Mtb H37Rv infection. Scale bar = 2.5 mm. (**D**) Analysis of percentage of inflamed area from each mouse lung (n=5 per group). The percentage inflamed area was evaluated using ImageJ software (NIH) and plotted as whisker box-plots (whiskers represent minimum and maximum values) and a student *t*-test was used for statistical analysis. Lung sections were derived from 5 mice per group (n=5) from *Figure 4a* experiments. Statistical analysis was conducted using student *t*-test. The error bars represent the standard deviation relative to the mean. *: p-value <0.05.

The online version of this article includes the following source data for figure 5:

**Source data 1.** Histopathological analysis of lung samples.

## Histopathological analysis of lung pathology after vaccination with rBCG::iE-DAP compared to WT BCG post Mtb challenge

As shown in *Figure 5a, b*, histopathological analysis of haematoxylin and eosin (H&E) stained lung samples from the vaccinated and Mtb challenged mice indicated that rBCG::iE-DAP+Dox immunized mice presented with early increased lung inflammation compared to WT BCG+Dox vaccinated mice. At 8 weeks post infection also, rBCG::iE-DAP+Dox immunized mice presented with increased inflamed sections of lung area compared to WT BCG+Dox immunized mice suggestive of sustained inflammation for control of infection (*Figure 5c, d*). The increased early inflammation in rBCG::iE-DAP+Dox immunized mice is reflective of early induction of anti-tuberculous immune responses, which were able to control growth early before establishment of infection and the sustained inflammation at 8 weeks post challenge is suggestive of enhanced immune responses during chronic disease stage which enable control of disease progression as shown in *Figure 4d*.

## Discussion

The BCG vaccine is given to children around the time of birth and is effective at preventing disseminated TB disease in young children (*Trunz et al., 2006*). However, BCG does not provide protection against TB infection in adults and has failed to eradicate the disease (*Dockrell and Butkeviciute, 2022*). This has spurred the need to develop novel TB vaccine candidates with varying modes of action to replace or boost BCG, which still remains the gold-standard for next generation TB vaccine development (*Martinez et al., 2022*). However, it is still unknown which mycobacterial antigens (either in Mtb or BCG) are able to induce effective protective anti-mycobacterial immunity. Also, there is evidence that BCG possesses several immune evasion mechanisms similar to those used by Mtb during infection to avoid immune killing that limit its efficacy as a vaccine (*Guinn and Rubin, 2017*). For example, rBCG strains further attenuated by deletion of immune evasion genes such as *sapM* (*Festjens et al., 2019*), *nuoG* (*Gengenbacher et al., 2016*), or *zmp1* (*Sander et al., 2015*) among others, have been developed and these show enhanced immunogenicity and efficacy against Mtb infection in animal models.

Immune evasion genes that are also essential for BCG viability are attractive targets to be studied for development of next generation rBCGs with enhanced efficacy. For example, genes encoding essential enzymes involved in the biosynthesis of potent immune-modulating cell wall lipids such as trehalose dimycolate (TDM), di- and tri-acylglycerols, pthiocerol dimycocerosates (PDIMs) and phenolic glycolipids (PGLs) are potential targets of gene regulation platforms to study their role in limiting BCG efficacy. Selective chemical removal of these lipids from BCG (i.e. delipidation of BCG) has shown the ability to enhance BCG efficacy in mice (*Moliva et al., 2019*). Gene regulation platforms, including CRISPRi are ideal platforms to study the effect of such essential immunomodulatory enzymes that can be targeted to enhance BCG efficacy. Indeed, recently a CRISPRi based rBCG (rBCG::CRISPRi-AftC) designed for the truncation of the anti-inflammatory cell wall associated lipoglycan – lipoarabinomannan (LAM) into the pro-inflammatory lipomannan derivative (LM) upon CRISPRi mediated depletion of the enzyme arabinofuranosyltransferase C (AftC, required for addition of D-arabinan branches on LM) was shown to enhance the immunogenicity of BCG by upregulating the expression of TNFα, a major pro-inflammatory cytokine (*Madduri et al., 2022*).

While previous approaches to modify BCG involved overexpression of protein antigens from Mtb in BCG to induce a long-lived type 1 helper T cell adaptive response, recent approaches that proved to be successful include improving the self-adjuvancy of BCG by re-engineering it to express innate immune cell activating antigens/adjuvants (*Angelidou et al., 2020*). These include for example overexpression of the STING agonist c-di-AMP in BCG or expression of the LTAK63 adjuvant in BCG which resulted in improved protection compared to standard BCG in a guinea pig model and a mouse model, respectively (*Carvalho Dos Santos et al., 2020*; *Dey et al., 2020*). We used the recently developed CRISPRi platform in BCG to target essential genes (*murT-gatD*) required for PG amidation implicated in immune evasion by masking the iE-DAP antigen, which has been shown to possess adjuvant potential (*Girardin et al., 2003*; *Moreno and Gatheral, 2013*). iE-DAP activates the NOD-1 PRR and several studies have shown the importance of this pathway in contributing to the onset of adaptive immunity (*Fritz et al., 2007*; *Mekonnen et al., 2018*). Moreover, NOD-2 has been shown to be the top upregulated gene in alveolar macrophages post subcutaneous vaccination with BCG in mice (*Mai et al., 2024*), thus we hypothesized that induction of NOD-1 activity would improve BCG efficacy.

Phenotypic characterization of rBCG::iE-DAP post CRISPRi activation, with SEM and TEM displayed changes in the outer cell wall surface in rBCG::iE-DAP respectively, consistent with the essentiality of MurT-GatD for PG crosslinking (*Shaku et al., 2023*). These defects were also associated with increased sensitivity to cell wall targeting antibiotics, confirming the essentiality of MurT-GatD mediated amidation of PG fragments for cell wall biosynthesis as previously described in other bacterial species (*Münch et al., 2012*; *Liu et al., 2017*; *Maitra et al., 2021*; *Shaku et al., 2023*). As PG amidation by the MurT-GatD complex is required for PG cross-linking by L,D-transpeptidases in mycobacteria (*Ngadjeua et al., 2018*), we further analysed the level of PG cross-linking in rBCG::iE-DAP by labeling the cells with BODIPY-FL vancomycin, a fluorescent vancomycin derivative binding uncrosslinked nascent PG monomers (*Miao et al., 2020*). We found that transcriptional repression of MurT-GatD expression in rBCG::iE-DAP was associated with complete cell wall labeling with this probe indicative of reduced PG cross-linking in rBCG::iE-DAP, due to lack of MurT-GatD enzymatic activity.

To probe specifically for the reduction of PG amidation in rBCG::iE-DAP, we used a previously developed amidation reporter probe, TetraFl (*Pidgeon et al., 2019*), to label MurT-GatD depleted cells and this showed increased labeling in rBCG::iE-DAP, indicative of reduced amidation upon transcriptional repression of MurT-GatD expression. Mechanistically, this modification results in reduced growth of mycobacteria (*Shaku et al., 2023*), and could also allow the recombinant BCG strain to persist during vaccination – a phenotype which has been suggested to enhance antigen presentation by BCG (*Kaveh et al., 2014*). As rBCG::iE-DAP is designed to activate the NOD-1 PRR by expression of the iE-DAP ligand, we performed qPCR analysis of *nod-1* expression in macrophages infected with rBCG::iE-DAP. We also assessed *nod-2* expression as mycobacteria also activate NOD-2 with *N*-glycolylated PG fragments (*Coulombe et al., 2009*). rBCG::iE-DAP induced substantially increased expression of both *nod-1* and *nod-2* in IFNγ activated macrophages and only significantly increased *nod-1* expression in unactivated macrophages. As *murT-gatD* depletion also causes cell wall defects, phagosomal killing of rBCG::iE-DAP could be causing efficient delivery of ligands such as iE-DAP to activate cytoplasmic PRRs like the NOD receptors. This indicated that while BCG is attenuated, further increasing its self-adjuvancy by activating expression of cell wall associated PRR ligands could be an ideal strategy to improve its efficacy.

We first tested rBCG::iE-DAP in an *in vitro* monocyte training assay (*Bekkering et al., 2016*) to assess the efficacy of this strain in training innate responses of macrophages. BCG induces a NOD-2-dependent trained immunity in monocytes resulting in epigenetic and metabolic reprogramming of monocytes which differentiate into macrophages with increased bactericidal properties (*Kleinnijenhuis et al., 2012*; *Blok et al., 2015*). We therefore tested rBCG::iE-DAP trained macrophages for their *in vitro* Mtb killing ability and found that in contrast to WT BCG trained macrophages, rBCG::iE-DAP trained macrophages displayed enhanced Mtb killing ability as measured by CFU counts 24 hr post infection. These results show that targeting innate immune responses with a modified live attenuated vaccine such as the rBCG::iE-DAP strain would enhance development of anti-mycobacterial immunity. We further show that rBCG::iE-DAP is responsive to activation in BMDMs using ATc and also in a mouse aerosol infection model using a minimal Dox dose, as Dox was previously shown to have immunomodulatory effects (*Miow et al., 2021*). Activation of rBCG::iE-DAP in BMDMs resulted in a trend of increased TNFα expression measured by ELISA providing evidence that expression of iE-DAP in rBCG::iE-DAP could enhance immunogenecity of BCG. These results bolstered our enthusiasm for investigation of rBCG::iE-DAP *in vivo* as a potential TB vaccine candidate.

We demonstrated that intradermal vaccination of mice with rBCG::iE-DAP, followed by administration of Dox for 6 weeks for CRISPRi mediated repression of MurT-GatD expression, resulted in superior protection from Mtb challenge in the lungs of the immunized mice compared to the WT BCG vaccine. Our vaccination experiments also included a WT BCG+Dox control group to rule out the role of Dox mediated immunomodulatory effects in enhancing WT BCG vaccine efficacy post Mtb challenge. Administration of Dox to WT BCG vaccinated mice (i.e. WT BCG+Dox group) did not enhance WT BCG vaccine efficacy against Mtb challenge when compared to the WT BCG without Dox control group, while rBCG::iE-DAP+Dox shows increased protection at both 4 and 8 weeks post Mtb challenge. We also observed a waning efficacy of the WT BCG vaccine at week 8 post Mtb infection, which has been previously reported (*Henao-Tamayo et al., 2015*; *Dwivedi et al., 2022*), however, interestingly vaccination with rBCG::iE-DAP+Dox remained effective at this time point.

Histopathological analysis of lung sections from the immunized and Mtb challenged mice shows that rBCG::iE-DAP+Dox induces early immune infiltration to the lung compared to WT BCG+Dox and this is maintained at least until 8 weeks post infection, providing early and sustained protection against Mtb challenge. Although increased inflammation in the lung could be detrimental to control of TB disease at a later stage, it has been suggested that an early balanced induction of pro-inflammatory and anti-inflammatory responses is required for optimal protection against Mtb infection (*Moreira-Teixeira et al., 2018*). Indeed, this was observed during host directed immunotherapy inducing early immune infiltration to the lung and this was correlated with improved protection against TB in a murine model (*Gress et al., 2023*). The immune correlates of protection induced by rBCG::iE-DAP are the subject of our future studies and will indicate whether increased inflammation at early time points of infection is important for vaccine mediated protection. As a CRISPRi based knockdown strategy was used in this study to create rBCG::iE-DAP, next steps will include construction of gene knockout mutants of rBCG::iE-DAP to generate a strain that is not based on CRISPRi as a TB vaccine candidate.

Collectively, our work demonstrates that MurT-GatD can be targeted in BCG to develop a new TB vaccine candidate.

# Materials and methods

## Key resources table

| Reagent type (species) or resource | Designation | Source or reference | Identifiers | Additional information |
|---|---|---|---|---|
| strain, strain background (*Mycobacterium bovis* BCG pasteur) | WT BCG | Gift from Dr Peter Sander (Institute of Medical Microbiology, University of Zurich, Zurich, Switzerland) | BCG Pasteur SmR | |
| strain, strain background (recombinant BCG::iE-DAP) | rBCG::iE-DAP | This paper | rBCG::iE-DAP | |
| strain, strain background (*Mycobacterium tuberculosis* H37Rv) | Mtb H37RvS | Centre of Excellence for Biomedical TB Research (Wits University, Johannesburg) | Mtb H37RvS | |
| strain, strain background (*Mycobacterium tuberculosis* H37Rv) | Mtb H37Rv | Center for Tuberculosis Research (Johns Hopkins University School of Medicine) | Mtb H37Rv | |
| cell line (THP-1 monocytes) | THP-1 monocytes | Gift from Dr Janine Scholefield (Council for Scientific and Industrial Research, South Africa) | | Cells authenticated by the supplier by morphology. Mycoplasma contamination test: negative |
| cell line (U937 promonocytes) | U937 monocytes | Gift from Dr Janine Scholefield (Council for Scientific and Industrial Research, Pretoria, South Africa) | | Cells authenticated by the supplier by morphology. Mycoplasma contamination test: negative |
| strain, strain background (BALB/c mice) | BALB/c mice | The Jackson laboratory (USA) | BALB/cJ strain #000651 | RRID:IMSR_JAX:000651 |
| strain, strain background (BALB/c SCID mice) | BALB/c SCID mice | The Jackson laboratory (USA) | CBySmn.Cg-Prkdc$^{scid}$/J strain#: 001803 | RRID:IMSR_JAX:001803 |

## Bacterial strains and culture conditions

### Growth conditions for *E. coli* DH5α and derivative strains

*E. coli* DH5α and derivative strains were grown in Luria-Bertani broth (LB) or on Luria-Bertani agar (LA) at 37°C with supplementation of the media with appropriate antibiotics. The antibiotic concentration used was as follows: Kanamycin (Kan): 50 µg/ml. Liquid cultures were grown at 37°C with shaking at a 100 rpm.

### Growth conditions for Mycobacterial and derivative strains

*M. bovis* BCG, *M tuberculosis* H37Rv and the recombinant BCG::iE-DAP strain were grown at 37 °C in Middlebrook 7H9 broth supplemented with OADC enrichment, 0.5% glycerol, 0.05% Tween 80 and appropriate antibiotics (hereafter referred to as Middlebrook 7H9 broth) or on Middlebrook 7H11 agar supplemented with OADC enrichment and 0.5% glycerol and appropriate antibiotics. The antibiotic concentration used for kanamycin was 50 µg/ml.

### Construction of rBCG::iE-DAP

The programmable mycobacterial CRISPRi system for repression of gene transcription was used as previously described by *Rock et al., 2017*, to generate the recombinant BCG::CRISPRi strain – rBCG::iE-DAP. Briefly, the CRISPRi system utilizes a catalytically inactivated *anhydrotetracycline*/doxycycline (ATc/Dox)-inducible CRISPRi dcas9 from *Streptococcus thermophiles*, which is directed by a (ATc/Dox)-inducible short-guide RNA (sgRNA) to specific target genes to prevent transcription initiation or elongation (*Rock et al., 2017*). sgRNAs were designed with the CRISPRi sgRNA design tool - https://pebble.rockefeller.edu/. The sgRNA sequence (top and bottom oligos) were annealed and cloned into BsmBI-digested CRISPRi vector PLJR965. These plasmids were introduced into *M. bovis* BCG by electroporation.

## Quantitative real-time PCR (qPCR) to assess *murT-gatD* transcriptional silencing

RNA was extracted using the Macherry-Nagel RNA extraction kit as per manufacturer's instructions and cDNA was prepared using the SuperScript IV reverse transcriptase (Invitrogen) as per manufacturer's instructions. Briefly, a 25 µl reaction was set up using 12.5 µl of a 2.5 µM reverse primer mix (Mb3739Rev: gattcaccgagcctggcag' and SigARev: cgcgcaggacctgtgagcgg) annealed to RNA sample, 4 µl 25 mM $MgCl_2$, 5 µl 5×first strand buffer, 2 µl 0.1 M DTT, 1 µl 10 mM dNTPs and 0.8 µl SuperScript III. PCR reactions were performed using the following parameters for reverse primer annealing: 94 °C for 90 s, 65 °C for 3 min and 57 °C for 3 min. cDNA synthesis was carried out using the following parameters: 50 °C for 5 min and 85 °C for 5 min. qPCR was performed using Sso Fast Evagreen Supermix (Bio-Rad) as per manufacturer's instructions. Briefly, 20 µl reactions were set up, each containing 10 µl Sso Fast Evagreen Supermix, 0.75 µl forward primer (Mb3739Fwd: gtcaaacgattcggtcagctg, SigAFwd: tgcagtcggtgctggacac) (10 µM), 0.75 µl reverse primer (Mb3739Rev: gattcaccgagcctggcag, SigARev: cgcgcaggacctgtgagcgg) (10 µM), 2 µl cDNA and nuclease-free water. All reactions were incubated in the CFX96 Real-Time PCR detection system (Bio-Rad) using the following parameters: 98 °C for 2 min followed by 39 cycles consisting of three steps – 98 °C for 5 s, 60 °C for 5 s and 72 °C for 5 s with SYBR Green quantification at the end of each cycle. Melt curve analysis was conducted from 65 °C with a gradual increase in 0.5 °C increments every 0.05 s to 95 °C with SYBR Green quantification conducted continuously throughout this stage. The raw data was analyzed using the Biorad CFX Manager 3.0 Software (Bio-Rad).

## Quantitative real-time PCR (qPCR) to assess *nod-1* and *nod-2* expression

RNA was extracted using the Macherry-Nagel RNA extraction kit as per manufacturer's instructions and cDNA was prepared using the SuperScript IV reverse transcriptase (Invitrogen) as per manufacturer's instructions. Briefly, 25 µl reactions were set up, each containing 2.5 µl of a 70 µM oligo d(T)$_{23}$, 4 µl 25 mM $MgCl_2$, 1 ug RNA, 5 µl 5×first strand buffer, 2 µl 0.1 M DTT, 1 µl 10 mM dNTPs and 0.8 µl SuperScript III and RNAse-free water to make up the volume. PCR reactions were performed using the following parameters: 94 °C for 90 s, 65 °C for 10 min and 57 °C for 3 min. qPCR was performed using Brilliant III Ultra-Fast SYBR green qPCR master mix (Agilent) as per manufacturer's instructions. Briefly, 20 µl reactions were set up, each containing 10 µl Brilliant III Ultra-Fast SYBR green qPCR master mix, 0.75 µl forward primer (NOD-1Fwd: caacggcatctccacagaagga, NOD-2Fwd: gcactgatgctg gcaaagaacg, GAPDHFwd: gtctcctctgacttcaacagcg) (10 µM), 0.75 reverse primer (NOD-1Rev: ccaa actctctgccacttcatcg, NOD-2Rev: cttcagtccttctgcgagagaac, GAPDHRev: accaccctgttgctgtagccaa), 2 µl cDNA and nuclease-free water. All reactions were incubated in the CFX96 Real-Time PCR detection system (Bio-Rad) using the following parameters: 98 °C for 2 min followed by 40 cycles consisting of three steps – 98 °C for 5 s, 60 °C for 5 s and 72 °C for 5 s with SYBR Green quantification at the end of each cycle. Melt curve analysis was conducted from 65 °C with a gradual increase in 0.5 °C increments every 0.05 s to 95 °C with SYBR Green quantification conducted continuously throughout this stage. The raw data was analyzed using the Biorad CFX Manager 3.0 Software (BioRad).

## Scanning electron microscopy (SEM) and transmission electron microscopy (TEM)

SEM and TEM were used to study the cell surface morphologies of the WT BCG and rBCG strains. The bacteria were immobilized to poly-l-lysine charged coverslips for 30 min and processed for SEM. Similarly, for TEM, bacterial suspensions were fixed and embedded in Spurr's resin. The immobilized bacteria were rinsed with phosphate buffered saline (PBS), and fixed in 2.0% paraformaldehyde, 2.0% glutaraldehyde in 1×PBS with 3 mM $MgC_{l2}$, pH 7.2 for 1 hr at room temperature. This was followed by 3 cycles of 10 min washes in sodium cacodylate buffer with 3% sucrose, samples were post-fixed in 0.8% potassium ferrocyanide, 1% $OsO_4$ and 3 mM $CaCl_2$ in 0.1 M sodium for 1 hr on ice in the dark. Samples were then rinsed in sodium cacodylate buffer and slowly rocked at 4 °C overnight. After a brief water rinse (2×5 min), bacteria were placed in 2% uranyl acetate for 1 hr at room tempera-ture in the dark. The samples were dehydrated through a graded series of ethanol to 100% EtOH, then a 1:1 solution of ethanol:Hexamethyldisiloxazne (HMDS) (Polysciences) followed by pure HMDS. Coverslips were dried in a desiccator overnight and then attached to aluminum stubs via carbon

sticky tabs (TedPella Inc), and coated with 20 nm of AuPd with a Denton Vacuum Desk III sputter coater. Stubs were viewed and digital images captured on a Leo 1530 field emission SEM operating at 1 kV. For TEM, equal volumes of 2×fixative (as described above) were added to bacterial suspensions and rocked for 10 min at room temperature. Samples were centrifuged, supernatant removed and 1×fixative added carefully to not disturb the pellet. All subsequent steps were identical to the protocol described above up for SEM to the final 100% ethanol step. Bacterial cells were transferred to propylene oxide, and gradually infiltrated with Spurr's low viscosity resin (Polysciences): propylene oxide. After 3 changes in 100% Spurr's resin, pellets were cured at 60 °C for 2 days. Sections were cut on a Reichert Ultra cut E with a Diatome Diamond knife. Eighty nm sections were picked up on formvar coated 1×2 mm copper slot grids and stained with tannic acid and uranyl acetate followed by lead citrate. Grids were viewed on a Phillips CM 120 TEM operating at 80 kV and digital images captured with an AMT 8 K x 8 K CCD camera.

## Fluorescent BODIPY-FL vancomycin staining

The Fluorescent BODIPY-FL vancomycin stain (Life Technologies) was used according to the manufacturer's instructions for analysis of PG synthesis in the mutant strains in comparison to the wildtype and complemented strains. The fluorescent vancomycin stain binds to the terminal dipeptide D-alanine-D alanine found on the PG stem peptide periplasmic precursor lipid II and consequently indicates the sites of new PG synthesis. The bacterial strains were grown to an OD600nm of 0.6 at 37 °C with shaking at a 100 rpm in 5 ml of Middlebrook 7H9 broth supplemented with appropriate antibiotics when necessary. Subsequently, 2 ml of the cells were harvested by centrifugation at 12 470×g for 5 min and the supernatant was discarded followed by washing of the cells with 500 µl of 0.01 M phosphate buffered saline (PBS), pH 7.4 and subsequent resuspension in 500 µl of 0.01 M PBS. Thereafter, 1.25 µl of vancomycin (200 micrograms/ml) and 2.5 µl of fluorescent BODIPY-FL vancomycin (100 micrograms/ml) were added to the cells followed by incubation at 37 °C with shaking for 1.5 hr. Following this, 500 µl of 0.01 M PBS was used to wash the cells three times, the cells were then resuspended in a 100 µl of 0.01 M PBS. For visualization, 5 µl of the cells was spotted on glass slides with 2% agarose pads. The slides were visualized with the Zeiss Observer Z1 inverted fluorescence microscope and the images taken were analyzed with the ZEN lite software (Zeiss) and Fiji software (ImageJ).

## Peptidoglycan extraction and labelling with an amine reactive dye

PG was extracted as previously described (*Shaku et al., 2023*). Briefly, wildtype BCG and rBCG::iE-DAP was grown to $OD_{600nm}$ of 2 and the cells were then harvested by centrifugation at 3500 × *g* for 10 min and resuspended in phosphate-buffered saline (PBS, pH 7.2). The cells were then lysed with a French press (Constant Systems). Insoluble material was obtained by centrifugation at 4000 × *g* for 30 min. The pellet was then resuspended in PBS containing 2% SDS and incubated at room temperature for 1 hr, then in PBS containing 2 mg/ml proteinase K and 2% SDS at 37 °C for 24 hours and finally in PBS containing 2% SDS at 90 °C for 1 hr. The extracted cell wall material was lyophilized, weighed and a 100 µg/ml of lyophilized cell wall material was resuspended in PBS and digested with 0.1 mg/ml mutanolysin for 24 hr. The digested material was harvested at 13,000 × *g* for 3 min, washed thrice with PBS. The pellet was resuspended in 500 µl PBS and labelled with 100 µg/ml of Alexa Fluor 488 NHS Ester (Sigma-Aldrich) for 3 hr. The CytoFLEX flow cytometer (Beckman Coulter) was used for analysis of the labelled PG samples (100 µl per sample) in the FITC channel (excitation/emission maxima = 494/517 nm). Three independent biological repeats were assessed.

## Flow cytometry

Flow cytometry was used for analysis of fluorophore labeled cells. Cells were grown in 5 ml of Middlebrook 7H9 broth supplemented with appropriate antibiotics at 37 °C with shaking to an $OD_{600nm}$ of 0.6. Thereafter, 1 ml of the culture was labelled with TetraFl (TAMRA-L-Ala-D-glutamine-L-Lys-D-Ala) for 3 hr at 37 °C with shaking. The CytoFLEX flow cytometer (Beckman Coulter) was used for analysis of TetraFl labeling.

## Mammalian cell culture

For cell-based *ex vivo* infection assays, the human monocyte U937 and THP-1 cell lines (obtained as a gift from the Council for Scientific and Industrial Research of South Africa [CSIR]) were grown in

RPMI-Glutamax (Cat. 61870–036, Fischer Scientific) supplemented with 10% heat inactivated fetal bovine serum (FBS) (Cat. 10082147, Fischer Scientific) at 37 °C with 5% $CO_2$. The cell lines (U937 and THP-1 monocytes) were authenticated by the manufacturer. The cell lines (U937 and THP-1 monocytes) were tested for mycoplasma contamination using the LookOut Mycoplasma PCR detection kit (Sigma-Aldrich) and both cell lines tested negative. BMDMs extracted from the bone marrow (BM) of 6–8 weeks old female wildtype BALB/c mice were cultivated in a similar manner. BMDMs were generated as previously described by *Toda et al., 2021*. Briefly, for differentiation of BM cells into macrophages, BM cells were seeded in BMDM differentiation media (RPMI-Glutamax supplemented with 10% FBS and 10% L929-conditioned media) and differentiated for 6 days. Non-adherent cells were washed out with warm BMDM differentiation media and adherent macrophages were used for *in vitro* infection assays.

## Mtb containment following *in vitro* training with BCG, rBCG or other antigens in human monocytic U937 cell lines

*In vitro* training of monocytes was performed according to a published model and Pan et al. (*Bekkering et al., 2016*; *Pan et al., 2020*). Briefly, U937 monocytes (1 × 10^6 /mL) were transferred into a 24-well plate and cells were incubated with either culture medium only as a negative control or MDP, LPS, heat killed WT BCG or heat killed rBCG::iE-DAP at 37 °C, and 5% $CO_2$ for 24 hr. Cells were washed twice with 1 mL of warm PBS and then incubated for 2 days in RPMI with 10% FBS and penicillin-streptomycin in the presence of 25 nM phorbol 12-myristate 13-acetate (PMA) (which can induce the differentiation of monocytes to macrophages). After washing twice with 1 mL of warm PBS, the differentiated macrophages were infected with Mtb H37Rv at MOI:1 and incubated for 24 hr. After 24 hr, cells were lysed and bacterial load was enumerated by plating for CFU counts on 7H11 Middlebrook media.

## Enzyme-linked immunosorbent assay (ELISA)

Sandwiched ELISA was performed for cytokine (TNF-α) measurement in culture supernatants. Culture supernatants were used immediately after harvest for ELISA. Sandwiched ELISA (R&D systems) was performed as per manufacturer's recommendations.

## BCG infection of BALB/c mice and CFU enumeration

To determine the lung bacillary burden of wild-type and rBCG::iE-DAP strains 6–8 weeks-old female BALB/c mice were infected using the aerosol route in a Glascol inhalation exposure system (Glasscol). Similarly, 6–8 weeks-old female BALB/c SCID mice were infected using the aerosol route as low dose aerosol infections with WT BCG lead to mouse lethality with a comparable time-to-death and offers highly uniform CFU lung implantations for each mouse (*Um et al., 2023*). The inoculum implanted in the lungs at day 1 (n = 3 mice per group) in female BALB/c mice was determined by plating the whole-lung homogenate on 7H11-selective plates containing carbenicillin (50 mg/ml), Trimethoprim (20 mg/ml), Polymyxin B (25 mg/ml) and Cycloheximide (10 mg/ml). Doxycycline was administered at determined doses for CRISPRi activation by daily oral gavage and following infection, mice lungs were harvested (n = 5 animals/group), homogenized in sterile PBS and plated on 7H11-selective plates at different dilutions. The 7H11-selective plates were incubated at 37 °C and single colonies were enumerated after 4 weeks for the 10 days aerosol infection experiment, and also after 4 weeks for the 8 weeks aerosol infection experiment.

## Mouse immunization and determination of protective efficacy against Mtb infection

Animal studies were performed as per the guidelines prescribed by the animal care and use committee of the Johns Hopkins University School of Medicine (protocol number: MO20M20). To test the efficacy of rBCG::iE-DAP as a vaccine candidate, BALB/c mice (n=10 per group) were immunized intradermally with 10^5 colony-forming units (CFU)/100 µL of WT BCG or rBCG::iE-DAP strains. Mice were sham immunized with saline (n=10) and Dox was administered by daily oral gavage to the Saline +Dox (n=5), WT BCG +Dox (n=5) and the rBCG::iE-DAP +Dox (n=5) groups for 6 weeks. Mice were weighed every week to monitor the effect of Dox administration on the health of the mice. Mice were challenged with ~100 CFU of Mtb H37Rv strain by the aerosol route 6 weeks post immunization in a Glasscol

inhalation exposure system (Glasscol). Lungs and spleens from infected animals were harvested at week 4 and week 8 post Mtb infection for analysis of lung bacterial burden by plating the whole-lung homogenate on 7H11-selective plates containing carbenicillin (50 mg/ml), Trimethoprim (20 mg/ml), Polymyxin B (25 mg/ml), and Cycloheximide (10 mg/ml) and lung pathology was assessed after hematoxylin and eosin (H&E) staining.

## Histopathology

Half of the left lung/mouse was cut and fixed in 10% neutral buffered formalin, paraffin embedded, sectioned, and H&E stained. Slides were digitally scanned (Aperio AT turbo scanner console version 102.0.7.5; Leica Biosystems, Vista, CA), transferred (Concentriq for Research version 2.2.4; Proscia, Philadelphia, PA), and visualized (Aperio ImageScope version 12.4.0.5043; Leica Biosystems Pathology Imaging, Buffalo Grove, IL). Histology images were analyzed with the Fiji software (ImageJ version 1.47 n [NIH]).

## Acknowledgements

We thank the SAMRC, Wits University research office and the Fulbright Scholarship program for providing MTS with scholarships to pursue this work. We thank Jeremy Rock (Rockefeller University) for donating CRISPRi plasmids and Peter Sander (University of Zurich) for the BCG Pasteur strain. We thank Dr Janine Scholefield for providing the U937 and THP-1 monocytes. We thank members of the CBTBR and Bishai Lab for helpful discussions and input on the manuscript. We are grateful for the assistance by Barbara Smith (Kuo Microscope Facility), Johns Hopkins School of Medicine for performing SEM and TEM and the Sidney Kimmel Comprehensive Cancer Center, Johns Hopkins School of Medicine for performing histology of lung samples. This work was supported by funding from an International Early Career Scientist Award from the Howard Hughes Medical Institute (to BDK), the South African National Research Foundation (to BDK), the South African Medical Research Council (to BDK, MS), the Centre for Aids Prevention Research in South Africa (CAPRISA, to BDK). This work was also supported by funding from NIH AI 155346.

## Additional information

### Competing interests

Bavesh D Kana: Senior editor, *eLife*. The other authors declare that no competing interests exist.

### Funding

| Funder | Grant reference number | Author |
| --- | --- | --- |
| Howard Hughes Medical Institute | HHMI000 | Bavesh D Kana |
| South African Medical Research Council | | Moagi Tube Shaku Bavesh D Kana |
| National Research Foundation | | Bavesh D Kana |
| National Institutes of Health | NIH AI 155346 | William R Bishai |

The funders had no role in study design, data collection and interpretation, or the decision to submit the work for publication.

### Author contributions

Moagi Tube Shaku, Conceptualization, Formal analysis, Investigation, Methodology, Writing – original draft, Writing – review and editing; Peter K Um, Formal analysis, Supervision, Investigation, Methodology; Karl L Ocius, Alexis J Apostolos, Marcos M Pires, Resources; William R Bishai, Resources, Data curation, Supervision, Methodology, Writing – review and editing; Bavesh D Kana, Conceptualization, Resources, Data curation, Formal analysis, Supervision, Funding acquisition,

Investigation, Methodology, Writing – original draft, Project administration, Writing – review and editing

### Author ORCIDs
Moagi Tube Shaku (iD) http://orcid.org/0000-0002-1171-7950
Peter K Um (iD) http://orcid.org/0000-0001-8215-9493
William R Bishai (iD) http://orcid.org/0000-0002-8734-4118
Bavesh D Kana (iD) http://orcid.org/0000-0001-9713-3480

### Ethics
All animal experiments were approved by the Johns Hopkins University Animal Care and Use Committee (Protocol number: MO20M20).

### Decision letter and Author response
Decision letter https://doi.org/10.7554/eLife.89157.sa1
Author response https://doi.org/10.7554/eLife.89157.sa2

## Additional files

### Supplementary files
• MDAR checklist

### Data availability
All data generated or analysed during this study are included in the manuscript and supporting file; Source data files have been provided for Figures 1, 2, 3, 4 and 5. Source data files are also provided for the supplementary information.

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
