## [Editor Report]

This important study provides evidence for a new target to improve vaccination against tuberculosis. The authors provide compelling evidence that inactivation of an essential enzyme pair in Mycobacterium bovis BCG, the only licensed vaccine against tuberculosis, enhances protection in a mouse model of the disease. The work will be of interest to researchers working on tuberculosis vaccine development.

---

## [Decision Letter]

**Decision letter after peer review:**

Thank you for submitting your article "A modified BCG with depletion of enzymes associated with peptidoglycan amidation induces enhanced protection against tuberculosis in mice" for consideration by *eLife*. Your article has been reviewed by 3 peer reviewers, one of whom is a member of our Board of Reviewing Editors, and the evaluation has been overseen by Wendy Garrett as the Senior Editor.

Essential revisions (for the authors):

1. There are no data to show that any protection or attenuation observed is due to increased signaling through NOD-1, given how pleiotropic the knockdown effect appears to be. Without a specific assay for NOD-1 involvement in the better protection, the authors should be careful to not draw this conclusion (i.e. Line 175, etc).

2. Lines 191-192 is not supported by the data given that the differences are not significantly different.

3. Lines 179-181: "At day 3 bacterial containment was observed for all strains but was most prominent for rBCG::iE-DAP strains treated with ATc.". Although the data is fairly clear (assuming this is referring to figure 2A), a statistical comparison would be helpful to support the interpretation of which strain is better controlled (and to support statement that the difference at 3 days is significant in the figure legend).

4. The authors confirm that they can induce attenuation of rBCG::iE-DAP by infecting via aerosol, but the protection experiment is done in vivo with intradermal administration. The authors could test knock down following intradermal administration because if this is not occurring efficiently it could explain the minimal significant effects in vivo.

5. Lines 250-253: This sentence does not reflect the data in the manuscript, and it is important to add more information in terms of time point and comparator. Specifically, only at a single time point is the WT BCG Dox group lower in lung weight and only compared to saline Dox. The rBCG::iE-DAP Dox group also never looks higher in lung weight, and if saline + Dox is not the comparator, what is? This also relates to Lines 257-259, where then the authors state that the lung weight is less, which is true based on their data, but contradictory to the prior statement. Then in the histology, the authors comment that there is more inflammation in the lungs of rBCG::iE-DAP Dox mice at 8 wpi, which raises the question of what the decreased lung weight is actually reflective of. In general, the biggest trend in the animal data is that the WT BCG with Dox is an outlier in terms of lower inflammation. The authors should confirm that the animal experiments were performed twice to make sure this is reproducible.

6. Lines 257- These experiments do not specifically look at dissemination to the spleen, any differences observed in the spleen may be due to reduced replication/fitness in the spleen.

7. In the Introduction the authors claim that close to 100 million infants have been vaccinated with BCG. They need to clarify over which time-frame this has occurred, as BCG has been given to more than 4.5 billion people since 1921 (mainly children).

8. For all figures, please specify the number of replicate experiments (where it should be at least 2).

9. Figure 3 B: The weight change should be plotted over time to allow objective interpretation of vaccination on safety and weight loss for all strains.

10. Figure 4 B and D: '% of inflamed area' should be plotted on a linear y-axis (not logarithmic) to objectively interpret the level of inflammation between the rBCG strain and the other groups.

11. Expand the discussion to synthesize the new information in the context of the field. The authors could discuss how the new rBCG differs from other rBCG strains; how would it be better than other TB vaccine candidates; how would a system that relies on doxy administration be translatable to human use; why is NOD-1 activation beneficial; why would an increased inflammation be acceptable for clinical use, etc. The authors could also comment on whether loss of D-glutamate has a general self-adjuvanting effect towards Mtb or whether MurT-GatD is no longer active in the later stages of Mtb infection and if recombinant BCG also enhance responses for other bacterial or viral vaccines.

12. Does expression of dCas9 in BCG affect bacterial growth with or without human cells or in animals?

13. The TAMRA-labelling effect is rather modest or the MurT-GatD-depleted cells? How about muropeptide analyses? Can muroppetide analysis reveal changes in glutamate amidation?

*Reviewer #1 (Recommendations for the authors):*

The authors find that CRISPRi knockdown of murT-gatD causes rather dramatic cell wall defects, more accessible cell wall labeling, and results in attenuated growth in macrophages and mice. There is some data presented to support that the murT-gatD KD strain may be more protective in the animal model, but most comparisons made are not significant and some interpretations stated in the Results section do not reflect the data in the figures. It seems that the most important comparisons are between WT BCG+Dox and rBCG+Dox, and the manuscript would be clearer if this comparison was focused on specifically. As it is written, it is often difficult to understand what is being compared to what and for which time point, especially because the manuscript jumps around quite a bit between timepoints.

Other comments:

1. There is no data to show that any protection or attenuation observed is due to increased signaling through NOD-1 given how pleiotropic the knockdown effect appears to be. Without a specific assay for NOD-1 involvement in the better protection, the authors should be careful to not draw this conclusion (i.e. Line 175, etc).

2. Lines 191-192 are not supported by the data given that the differences are not significantly different.

3. Lines 179-181: "At day 3 bacterial containment was observed for all strains but was most prominent for rBCG::iE-DAP strains treated with ATc.". Although the data is fairly clear (assuming this is referring to figure 2A), a statistical comparison would be helpful to support the interpretation of which strain is better controlled (and to support the statement that the difference at 3 days is significant in the figure legend).

4. The authors confirm that they can induce attenuation of rBCG::iE-DAP by infecting via aerosol, but the protection experiment is done in vivo with intradermal administration. The authors could test knockdown following intradermal administration because if this is not occurring efficiently it could explain the minimal significant effects in vivo.

5. Lines 250-253: This sentence does not reflect the data in the manuscript, and it is important to add more information in terms of time points and comparator. Specifically, only at a single time point is the WT BCG Dox group lower in lung weight and only compared to saline Dox. The rBCG::iE-DAP Dox group also never looks higher in lung weight, and if saline + Dox is not the comparator, what is? This also relates to Lines 257-259, where then the authors state that the lung weight is less, which is true based on their data, but contradictory to the prior statement. Then in the histology, the authors comment that there is more inflammation in the lungs of rBCG::iE-DAP Dox mice at 8 wpi, which raises the question of what the decreased lung weight is actually reflective of. In general, the biggest trend in the animal data is that the WT BCG with Dox is an outlier in terms of lower inflammation. The authors should confirm that the animal experiments were performed twice to make sure this is reproducible.

6. Lines 257- These experiments do not specifically look at dissemination to the spleen, any differences observed in the spleen may be due to reduced replication/fitness in the spleen.

*Reviewer #2 (Recommendations for the authors):*

To strengthen the manuscript, the authors need to repeat the in vivo vaccination-challenge experiments to (i) show reproducibility; (ii) increase statistical power; and (iii) demonstrate that rBCG truly is more protective but more inflammatory.

In the Introduction the authors claim that close to 100 million infants have been vaccinated with BCG. They need to clarify over which time-frame this has occurred, as BCG has been given to more than 4.5 billion people since 1921 (mainly children).

Figure 3 B: The weight change should be plotted over time to allow objective interpretation of vaccination on safety and weight loss for all strains.

Figure 4 B and D: '% of inflamed area' should be plotted on a linear y-axis (not logarithmic) to objectively interpret the level of inflammation between the rBCG strain and the other groups.

The Discussion section of the paper reads like a repetition of the results. This whole section needs to be expanded and changed. At the very least, the discussion should discuss how the new rBCG differs from other rBCG strains; how would it be better than other TB vaccine candidates; how would a system that relies on doxy administration be translatable to human use; why is NOD-1 activation beneficial; why would an increased inflammation be acceptable for clinical use;….etc., etc. A discussion should be a critical reflection of the literature and how their results fit in with current dogmas, ideas and strategies.

*Reviewer #3 (Recommendations for the authors):*

Does expression of dCas9 in BCG affect bacterial growth with or without human cells or in animals?

The TAMRA-labelling effect is rather modest or the MurT-GatD-depleted cells? How about muropeptide analyses? Can muroppetide analysis reveal changes in glutamate amidation?

Discussion:

Can the authors comment on whether loss of D-glutamate has a general self-adjuvanting effect towards Mtb or whether MurT-GatD is no longer active in the later stages of Mtb infection? Would recombinant BCG also enhance responses for other bacterial or viral vaccines?

[Editors' note: further revisions were suggested prior to acceptance, as described below.]

Thank you for resubmitting your work entitled "A modified BCG with depletion of enzymes associated with peptidoglycan amidation induces enhanced protection against tuberculosis in mice" for further consideration by *eLife*. Your revised article has been evaluated by Wendy Garrett (Senior Editor) and a Reviewing Editor. We deeply apologize for the delay in returning the reviews, we had website related technical issues that stemmed from a switch to a new BRE.

The manuscript has been improved, but there are some remaining issues that need to be addressed. In particular, Reviewer #2 requested that you address their questions about the timing and methods used to obtain new data presented in figure 3 in the response letter. Please see below for details.

*Reviewer #2 (Recommendations for the authors):*

The authors have substantially revised the manuscript and have addressed all of the reviewer comments in great depth. As a consequence, the manuscript has substantially improved.

The only additional question I have relates to the inclusion of new data for Figure 3. Here, the authors have addressed the safety concern raised due to the enhanced inflammation by performing aerosol infections of SCID mice. They show that all groups of mice succumb roughly around the same time between 200-300 days.

Given that the first review of the manuscript occurred in June 2023, only approximately 8 months (240 days) have passed since, and it is unclear how a 300+ day experiment could have been performed in the meantime. This requires further explanation.

Furthermore, normally SCID mice safety experiments are performed via intravenous injection of about 1x10e6 bacteria and BCG Pasteur leads to a 100% mortality of SCID mice in about 50-100 days in that system. Why was the aerosol infection route chosen? Please also clarify.

---

## [Author Response]

Essential revisions (for the authors):1. There are no data to show that any protection or attenuation observed is due to increased signaling through NOD-1, given how pleiotropic the knockdown effect appears to be. Without a specific assay for NOD-1 involvement in the better protection, the authors should be careful to not draw this conclusion (i.e. Line 175, etc).

Experiment performed and added: We agree with the reviewer here, this is an important point. We tried to address this by assessing expression of the genes encoding the NOD1 receptor, in macrophages, in response to exposure to our recombinant BCG compared to the parental. We demonstrate substantive induction of the NOD1 receptor gene with the recombinant BCG compared to the parental strain. These data are provided in in figure 1j and figure S6b for macrophages infected with rBCG::iE-DAP showing substantially increased NOD-1 transcription after 12 hours post infection. These new data are reported in Line178-192. As this does not definitively demonstrate involvement of NOD1 induction during infection in mice, we also refined our statements in this respect to provide a more careful synthesis of the data.

2. Lines 191-192 is not supported by the data given that the differences are not significantly different.

Given that the rBCG only displayed an increasing trend of TNFα induction in activated BMDMs although this was not statistically significant – in line 209-214 (previously 191-192) we remove the phrase suggesting that rBCG signals for increased cytokine expression. Based on this, these data are in the supplementary information (Figure 2 supplement 1).

3. Lines 179-181: "At day 3 bacterial containment was observed for all strains but was most prominent for rBCG::iE-DAP strains treated with ATc.". Although the data is fairly clear (assuming this is referring to figure 2A), a statistical comparison would be helpful to support the interpretation of which strain is better controlled (and to support statement that the difference at 3 days is significant in the figure legend).

A statistical comparison was performed for day 3 CFU of WT BCG vs rBCG in BMDM infection experiments and figure 2A was updated accordingly. Although rBCG+ATc shows reduced survival at this time point compared to WT BCG, this was not statistically significant with p-values found to be as indicated below:

WT BCG vs rBCG 0 ng/ml ATc: p-value=0.947

WT BCG vs rBCG 100 ng/ml ATc: p-value = 0.308

WT BCG vs rBCG 200 ng/ml ATc: p-value = 0.401

WT BCG vs rBCG 300 ng/ml ATc: p-value = 0.294

WT BCG vs rBCG 400 ng/ml ATc: p-value = 0.309

WT BCG vs rBCG 100 ng/ml ATc: p-value = 0.243

We update the figure 2 legend to indicate that although reduced growth was observed at day 3, this was statistically significant only on day 5

4. The authors confirm that they can induce attenuation of rBCG::iE-DAP by infecting via aerosol, but the protection experiment is done in vivo with intradermal administration. The authors could test knock down following intradermal administration because if this is not occurring efficiently it could explain the minimal significant effects in vivo.

Testing knockdown following intradermal vaccination would not be feasible as this would entail excision of the vaccination site to isolate bacteria. To best of our knowledge, such an approach is not widely used in the field as the bacteria do not necessarily remain at the vaccination site after administration. Rather they are taken up by immune cells and can traffic to other parts of the body. This is in contrast to aerosol infection where, at least early during infection, the bacteria are primarily located in the lung, prior to dissemination elsewhere. Hence, we kept the duration of this experiment very short (10 days) where measured the ability to activate the strain. To show that recombinant BCG is stable over time in mice, we performed a long-term experiment (for durations of 4 and 8 weeks, as per our challenge model) where mice were infected with the recombinant strain and bacteria sampled over time to confirm presence of the CRISPR plasmid, as measured by a reduction of colony size in the presence of DOX and the presence of CRISPR-dCAS9 as detected by PCR. The data for week 8 are given in Figure 2I and the data for 4 weeks are given in Figure S8.

5. Lines 250-253: This sentence does not reflect the data in the manuscript, and it is important to add more information in terms of time point and comparator. Specifically, only at a single time point is the WT BCG Dox group lower in lung weight and only compared to saline Dox. The rBCG::iE-DAP Dox group also never looks higher in lung weight, and if saline + Dox is not the comparator, what is? This also relates to Lines 257-259, where then the authors state that the lung weight is less, which is true based on their data, but contradictory to the prior statement. Then in the histology, the authors comment that there is more inflammation in the lungs of rBCG::iE-DAP Dox mice at 8 wpi, which raises the question of what the decreased lung weight is actually reflective of. In general, the biggest trend in the animal data is that the WT BCG with Dox is an outlier in terms of lower inflammation. The authors should confirm that the animal experiments were performed twice to make sure this is reproducible.

We thank the reviewer for this comment. In line 283-286 (previously line 250-253), we rephrase the previous sentence to reflect the data in the manuscript. The reviewer is correct to state that only the WT BCG+Dox group only displays significantly reduced lung weights at week 4 postinfection in comparison to the saline+Dox group at this time point while the other groups are similar to the saline+dox group in lung weight. In line 293-295 (previously 257-259), we correctly reference the data presented in the manuscript to state that at week 8 the rBCG::iE-DAP+Dox group displayed reduced lung weights compared to the Saline+Dox group and the WT BCG vaccinated group. We conducted these experiments at two points, with 5 mice per group. This approach is standard in the field.

6. Lines 257- These experiments do not specifically look at dissemination to the spleen, any differences observed in the spleen may be due to reduced replication/fitness in the spleen.

The reviewer is correct. In line 291-294, previously line 257, we replace the phrase “reduced bacterial dissemination to the spleen” with “reduced bacterial burdens in the spleen” to indicate that the differences observed may be due to reduced replication/fitness in the spleen and not due to reduced dissemination.

7. In the Introduction the authors claim that close to 100 million infants have been vaccinated with BCG. They need to clarify over which time-frame this has occurred, as BCG has been given to more than 4.5 billion people since 1921 (mainly children).

In line 97, we rephrase the sentence to indicate that BCG is administered to close to 100 million infants annually.

8. For all figures, please specify the number of replicate experiments (where it should be at least 2).

In Figure 1 legend we indicate that “Three independent biological repeats were assessed per experiment”. In figure 2 legend we indicate that “Three independent biological repeats were assessed for the in vitro experiments. 5 mice per group (n=5) were used for the in vivo experiments”. In figure 3 legend we indicate that “5 mice per group (n=5) were used for the in vivo experiments”. In figure 4 legend we indicate that “5 mice per group (n=5) were used for the in vivo experiments”. In figure 5 legend we indicate that “Lung sections were derived from 5 mice per group (n=5) from Figure 4 experiments”. In Figure 1 supplement 2 legend we indicate that “Three independent biological repeats were assessed”. In Figure 1 supplement 3 legend we indicate that “Three independent biological repeats were assessed”. In Figure 1 supplement 4 legend we indicate that “Three independent biological repeats were assessed”. In Figure 1 supplement 5 legend we indicate that “Three independent biological repeats were assessed”. In Figure 2 supplement 1 legend we indicate that “Three independent biological repeats were assessed. In Figure 2 supplement 2 legend we indicate that “Three independent biological repeats were assessed” for the in vitro experiments and 5 mice per group (n=5) were used for the in vivo experiments”. In Figure 4 supplement 1 legend we indicate that Lung sections were derived from 5 mice per group (n=5) from Figure 4a experiments”.

9. Figure 3 B: The weight change should be plotted over time to allow objective interpretation of vaccination on safety and weight loss for all strains.

In Figure 4B (previously figure 3B) we plot the weight change over time to allow objective assessment /interpretation of vaccination and Doxycycline administration on safety and weight loss for all strains.

10. Figure 4 B and D: '% of inflamed area' should be plotted on a linear y-axis (not logarithmic) to objectively interpret the level of inflammation between the rBCG strain and the other groups.

In figure 5B and D (previously 4B and D) the % of inflamed area' is now plotted on a linear y-axis scale.

11. Expand the discussion to synthesize the new information in the context of the field. The authors could discuss how the new rBCG differs from other rBCG strains; how would it be better than other TB vaccine candidates; how would a system that relies on doxy administration be translatable to human use; why is NOD-1 activation beneficial; why would an increased inflammation be acceptable for clinical use, etc. The authors could also comment on whether loss of D-glutamate has a general self-adjuvanting effect towards Mtb or whether MurT-GatD is no longer active in the later stages of Mtb infection and if recombinant BCG also enhance responses for other bacterial or viral vaccines.

In line 319-323 and 369-377, we further expand on the Discussion section to indicate that it is still unknown exactly which antigens are able to induce protective anti-mycobacterial immunity and in line 340-350 we detail how rBCG::iE-DAP which expresses the NOD-1 activating antigen (iE-DAP) differs from other rBCG strains.

In line 386-388 we elaborate on how modifying BCG to express non-protein antigens such as the iE-DAP antigen in the cell wall to activate innate immune cells would enhance the development of anti-mycobacterial immunity by improving the self-adjuvancy of BCG as most TB vaccine candidates are designed to only induce T helper 1 responses which appear to be limited for protection.

We used a doxycycline/ATc dependent system to probe accessibility of the CRISPRi system in vivo, however, for translation to human use we’re developing a knockout of the MurT gene in BCG-Pasteur, however, the deletion of this gene is only possible when we recover transformants on Vitamin B12 as we have discovered that GatD also known as CobQ2 which is in an operon with MurT, plays a role in VitB12. We are currently generating knockout mutants of MurT and confirming the genetic details of this strain which will be eligible for human use. In line 423-425, we also indicate that the CRISPRi system only allowed us to probe this pathway however the next steps would be to generate MurT knockout strains and to study the correlates of protection induced by this modified Rbcg.

NOD-1 activation during infection is necessary for induction of anti-bacterial immune responses (PMID: 17433730) and several studies have demonstrated a role for NOD-1 mediated immune responses in defense against TB infection (PMID: 29868226). Our hypothesis was that activation of NOD-1 with a modified BCG during vaccination would enhance the immunogenicity of the rBCG strain. We provide new data in Figure 1i and 1j demonstrating increased nod-1 transcription in rBCG::iE-DAP infected macrophages and In line 354-356 we further elaborate on why activation of the NOD-1 pathway with a recombinant BCG strain would be beneficial for development of anti-myobacterial immunity.

We further provide data previously left out (now Figure 3) where we tested the attenuation or virulence of rBCG::iEDAP compared to wildtype BCG in SCID mice. This data shows that in an aerosol infection model of SCID mice, rBCG::iE-DAP+DOX is similarly attenuated as WT BCG and not causing increased death of the infected SCID mice. Although 8 weeks post-infection of vaccinated WT Balb/C mice with Mtb we observed slightly increased inflammation in the lungs of rBCG::iE-DAP+DOX vaccinated mice compared to WT BCG+DOX (Figure 4C, right panel), we observed no significant difference in WT BCG without DOX. This data suggests that the observed slight increase in inflammation could be representative of immune responses that allowed for bacterial clearance as observed in Figure 4C (right section). In line 416-419, we expand on the findings of the histopathology of the mouse lungs and indicate that “Although increased inflammation could be detrimental to control of disease, it has been suggested that a balanced induction of proinflammatory and anti-inflammatory responses is required for optimal protection against Mtb infection”.

In line 349-350, we further comment on the fact that this modification (i.e expression of the iE-DAP antigen) induces immune responses fine-tuned for a bacterial infection and as BCG possesses similar antigens to Mtb, this results in a general self-adjuvanting effect towards Mtb infection.

We do not expect MurT-GatD to be nonactive in later stages of Mtb infection. In the model organism M. smegmatis we have found that this enzyme pair is essential for viability (doi: 10.3389/fcimb.2023.1205829).

We cannot ruleout that increasing the self-adjuvancy of BCG would result in enhanced responses for other bacterial / viral infections as BCG has been shown to induce trained innate immune responses against non-specific bacterial or viral infections. However, without the evidence that rBCG::iE-DAP can induce such nonspecific immune responses we choose not to comment on this.

12. Does expression of dCas9 in BCG affect bacterial growth with or without human cells or in animals?

Expression of dCas9 (specifically the dCas9Sth1 version used in our study) in axenic culture of mycobacteria was shown to display minimal proteotoxicity ([PMID: 28165460]). A recent study (PMID: 36405350) also showed that dCas9 expression in BCG does not cause toxicity also in human macrophage cell line.

13. The TAMRA-labelling effect is rather modest or the MurT-GatD-depleted cells? How about muropeptide analyses? Can muroppetide analysis reveal changes in glutamate amidation?

Experiment performed and added: We thank the reviewer for this comment. We have struggled to establish an appropriate protocol for MS detection of the amide modification in mycobacterial PG. In our initial submission, we chose to use the synthetic probe (TetraFl-1 [PMID: 31487148]) to label MurT-GatD depleted cells as this is a non-invasive way to assess cell wall biochemistry. This probe has been chemically engineered to specifically interrogate the requirement for glutamate amidation at position 2 in the stem peptide. In earlier work from our lab (https://doi.org/10.3389/fcimb.2023.1205829), we used variations of this probe where position 2 of probe was modified to carry either a glutamate or a glutamine (which carries the amine modification). Using this approach, we were able to demonstrate that only the probe carrying glutamine was incorporated into mycobacterial PG. This confirmed the requirement for amidation at position 2 (on the receiving side-chain) for PG crosslinking. Hence, we hypothesized that if this amidation modification was reduced on the receiving side chain, the PG biosynthetic machinery will become more reliant on the donor peptide (in the fluorescent probe) for crosslinking and will hence incorporate more probe. We agree that this effect was marginal. To further confirm the reduction in amidation at position 2 of the receiving stem peptide in our recombinant BCG strain, we now assess the level of PG amidation by assessing labeling with an amine reactive dye. We previously demonstrated (https://doi.org/10.3389/fcimb.2023.1205829) that it reports on the level of PG amidation in bacteria. Labeling of PG extracted from WT BCG vs PG extracted from MurT-GatD depleted rBCG::iE-DAP revealed reduced labelling of the rBCG, thus confirming reduced amidation (see figure S5). These data are now reported in lines 175-178.

[Editors’ note: what follows is the authors’ response to the second round of review.]

The manuscript has been improved, but there are some remaining issues that need to be addressed. In particular, Reviewer #2 requested that you address their questions about the timing and methods used to obtain new data presented in figure 3 in the response letter. Please see below for details.Reviewer #2 (Recommendations for the authors):The authors have substantially revised the manuscript and have addressed all of the reviewer comments in great depth. As a consequence, the manuscript has substantially improved.The only additional question I have relates to the inclusion of new data for Figure 3. Here, the authors have addressed the safety concern raised due to the enhanced inflammation by performing aerosol infections of SCID mice. They show that all groups of mice succumb roughly around the same time between 200-300 days.Given that the first review of the manuscript occurred in June 2023, only approximately 8 months (240 days) have passed since, and it is unclear how a 300+ day experiment could have been performed in the meantime. This requires further explanation.

The reviewer is correct in the calculation of dates regards the SCID mouse experiment and time of manuscript submission. However, we had started the SCID mouse experiment on the 16th May 2022. The date of first submission of the manuscript was 4th May 2023. We could not include this experiment in the first submission as we did not have all the outcomes and final analysis at the date of submission. Given that the fellowship for the primary author was ending, we were pressed to submit the manuscript before this experiment was concluded and analysed. We were able to include the final results in the revision. All animal experiments were conducted at Johns Hopkins University, under the oversight of the prevailing Institutional Animal Care and Use Committee. If required, these details can be verified by the institution and animal husbandry staff.

Furthermore, normally SCID mice safety experiments are performed via intravenous injection of about 1x10e6 bacteria and BCG Pasteur leads to a 100% mortality of SCID mice in about 50-100 days in that system. Why was the aerosol infection route chosen? Please also clarify.

We agree with the reviewer that an IV infection with 1x10e6 BCG will typically lead to mouse death in 50-100 days. It is our experience that low dose aerosol infections of BCG lead to mouse lethality with a comparable time-to-death https://www.biorxiv.org/content/10.1101/2023.12.15.571740v1. We chose to use the aerosol route because in our experience it offers highly uniform CFU implantations for each mouse, it is less labour intensive than IV injections, and it avoids the hazard of sharps containing viable bacilli for laboratory staff. To address this, we have now added the relevant text to the revised manuscript at lines 594 to 596.